# CAPABILITY-BASED SCALING TRENDS FOR LLM-BASED RED-TEAMING

**Alexander Panfilov**[1,2,3]  **Paul Kassianik**[4]  **Maksym Andriushchenko**[5†]  **Jonas Geiping**[1,2,3†]

[1]ELLIS Institute Tübingen  [2]Max Planck Institute for Intelligent Systems  [3]Tübingen AI Center

[4]Foundation AI – Cisco Systems Inc.  [5]EPFL

alexander.panfilov@tue.ellis.eu

## ABSTRACT

As large language models grow in capability and agency, identifying vulnerabilities through red-teaming becomes vital for safe deployment. However, traditional prompt-engineering approaches may prove ineffective once red-teaming turns into a *weak-to-strong* problem, where target models surpass red-teamers in capabilities. To study this shift, we frame red-teaming through the lens of the *capability gap* between attacker and target. We evaluate more than 600 attacker-target pairs using LLM-based jailbreak attacks that mimic human red-teamers across diverse families, sizes, and capability levels. Three strong trends emerge: (i) more capable models are better attackers, (ii) attack success drops sharply once the targets capability exceeds the attacker's, and (iii) attack success rates correlate with high performance on social science splits of the MMLU-Pro benchmark. From these observations, we derive a *jailbreaking scaling curve* that predicts attack success for a fixed target based on attacker-target capability gap. These findings suggest that fixed-capability attackers (e.g., humans) may become ineffective against future models, increasingly capable open-source models amplify risks for existing systems, and model providers must accurately measure and control models' persuasive and manipulative abilities to limit their effectiveness as attackers.

## 1 INTRODUCTION

Large language models (LLMs) are rapidly evolving into powerful general-purpose systems, capable of reasoning (Guo et al., 2025), task completion (OpenAI, 2025), and even conducting research (Intology AI, 2025). Alongside this rise, substantial efforts have been made to ensure the *safety* of these models. As part of the pre-release safety evaluation process, human red-teamers often probe LLMs for failure modes and unsafe behaviors (Anthropic, 2024; Kavukcuoglu, 2025). This gives rise to various *jailbreaking attacks*, aimed at eliciting harmful behaviors in worst-case scenarios, i.e., assessing how *secure* or adversarially aligned a model is (Carlini et al., 2023; Qi et al., 2024a).

The real-world harm from jailbroken models remains rather limited (Geiping et al., 2024), if present at all (Willison, 2023). However, the core argument is that as general and agentic capabilities advance, sufficiently integrated AI systems will pose very practical security risks (Rando et al., 2025; Bostrom, 2014). Robey et al. (2025) offer a glimpse of such a future: an LLM-powered robot dog is jailbroken using a purely black-box RoboPAIR attack, leading to physical-world harm.

However, some foresee a future where AI systems become impossible to jailbreak (Kokotajlo et al., 2025). While Kokotajlo et al. (2025) offer no empirical evidence for such maximalist predictions, we observe two orthogonal trends that point in that direction for human-like black-box red-teaming: (i) safety mechanisms are getting stronger (both system-level (Sharma et al., 2025) and model-level (Zou et al., 2024; Kritz et al., 2025)); and (ii) models themselves are becoming smarter in general.

---

[†]Equal supervision.
Code available at https://github.com/kotekjedi/capability-based-scaling.

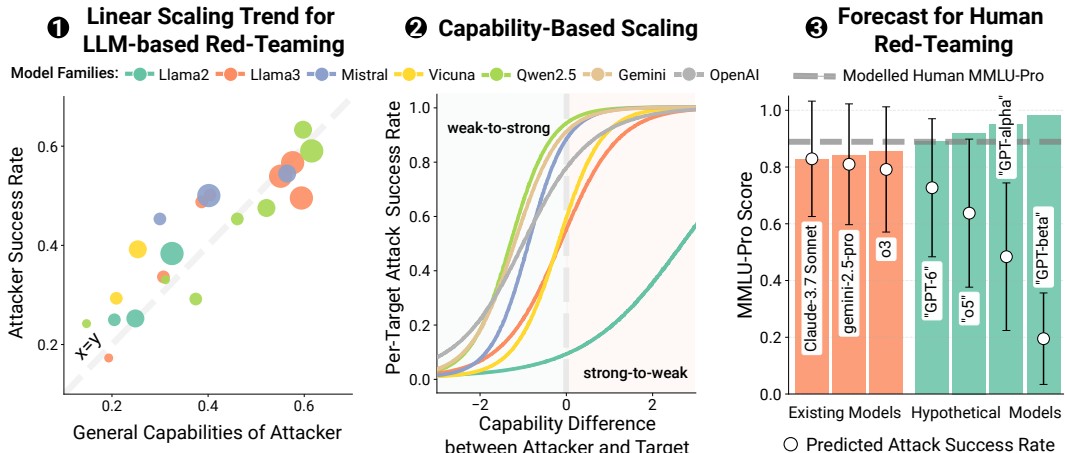

**Figure 1: Overview of Our Contributions.** ❶ We evaluate over 600 attacker-target combinations with four LLM-based jailbreak attacks and find that attacker success rate scales linearly with general capability (measured with MMLU-Pro scores). ❷ However, for a fixed target model the attack success rate follows a sigmoid curve and can be predicted accurately from the attacker-target capability gap. ❸ Using the scaling trends observed in currently deployed models, we forecast that red-teaming by any fixed-capability attacker, such as a human, will inevitably become less effective as target models' capabilities increase.

This increase in *capability* means that models are also better at adhering to safety guidelines and better at reasoning about user intent (Zaremba et al., 2025; Ren et al., 2024b).

As models become more capable, red-teaming is increasingly being cast as a *weak-to-strong* problem. This contrasts with the vast majority of current black-box attacks, which "outsmart" target models in a variety of ways: clever prompt engineering (Liu et al., 2023), role-playing (Shah et al., 2023), social-engineering techniques (Zeng et al., 2024), etc. In an attempt to understand how future weak-to-strong dynamics may impact model security, we ask:

> *At what capability gap might human-like red-teaming become infeasible?*

To answer this question, we model the success of red-teaming as a function of the *capability gap* (the difference in benchmark scores, e.g., MMLU-Pro (Wang et al., 2024)) between **Attacker** and **Target**. To evaluate capabilities on equal footing, we implement four human-like LLM-based jailbreaking attacks: PAIR (Chao et al., 2025), TAP (Mehrotra et al., 2024), PAP (Zeng et al., 2024) and Crescendo (Russinovich et al., 2025). We execute them on over 600 attacker-target pairs, examining 29 target models across a variety of families, parameter sizes, and capability levels (see Figure 2). We consider exclusively LLM-based red-teaming attacks. We apply *model unlocking* (Qi et al., 2024b; Volkov, 2024) to remove safety guardrails from open-source models while preserving their general capabilities for use as attackers (see Section 3.2 and Appendix A).

This large-scale study yields several key insights that contribute to our understanding of future black-box red-teaming. These are as follows:

- **Stronger Models are Better Attackers.** Attacker success, averaged over targets, rises almost linearly with general capability ($\rho > 0.88$; see Section 4). This underscores the need to benchmark models' *red-teaming* capabilities (as opposed to defensive capabilities) before release.

- **A Capability-Based Red-Teaming Scaling Curve.** For most currently released models, including closed-source ones, attack success rate (ASR) declines predictably as the capability gap between attackers and targets increases and can be *accurately modeled as a sigmoid function* (see Section 4 and Section 5). This finding suggests that while human red-teamers will become less effective against advanced models, increasingly capable open-source models will put existing LLM systems at risk.

- **Social-Science Capabilities are Stronger ASR Predictors than STEM Knowledge.** Model capabilities related to social sciences (e.g., health, psychology and philosophy) are more strongly correlated with attacker success rates than STEM capabilities (Section 6). This finding highlights the need to measure and control models' persuasive and manipulative capabilities.

Taken together, our findings offer a practical framework for reasoning about how long LLM-powered applications are likely to remain safe in the face of advancing LLM attackers. They underscore the

need for model providers to further invest in improving robustness, scalable automated red-teaming and systematic benchmarking of persuasion and manipulative abilities of models.

## 2 RELATED WORK

**Human Red-Teaming.** To prevent harmful behavior in deployed models, LLM providers employ manual red-teaming, where human testers attempt to elicit unsafe outputs and refine model responses through targeted feedback (Anthropic, 2024; Team Gemini et al., 2024; Ganguli et al., 2022). While effective for identifying certain behavioral flaws, this approach is not scalable: it relies on creativity, manual data curation, and high-cost human oversight. Moreover, human red-teamers often fail to discover unnatural but highly effective inputs that are uncovered by automated white-box jailbreak attacks (Zou et al., 2023; Andriushchenko et al., 2025). Nonetheless, some human-discovered strategies, such as multi-turn attacks (Li et al., 2024a), past-tense framing (Andriushchenko & Flammarion, 2025), and payload-splitting (Liu et al., 2023) do not emerge naturally from automated pipelines and show strong transfer across models once discovered.

**Automated Red-Teaming.** Automated red-teaming, or *jailbreaking*, has emerged as a scalable way to benchmark LLMs under worst-case safety scenarios (Chao et al., 2024; Mazeika et al., 2024; Perez et al., 2022) with attack success rate (ASR) as the primary evaluation metric. To emulate human-like probing strategies, numerous LLM-based jailbreak methods have been proposed (Chao et al., 2025; Mehrotra et al., 2024; Russinovich et al., 2025; Pavlova et al., 2024; Sabbaghi et al., 2025) where an attacker model is guided by a red-teaming prompt containing human-curated in-context demonstrations. These methods operate under a black-box threat model that mirrors human constraints and broadly reflect human-like strategies (Shah et al., 2023; Schulhoff et al., 2023; Li et al., 2024a; Zeng et al., 2024). These strategies include role-playing (Chao et al., 2025; Shen et al., 2024), word substitution (Chao et al., 2025), emotional appeal (Chao et al., 2025; Zeng et al., 2024), usage of the past tense (Russinovich et al., 2025), decomposing harmful queries over multiple turns (Glukhov et al., 2025; Russinovich et al., 2025), and others. While typically less effective than white-box algorithmic attacks (Boreiko et al., 2025), the most capable LLM-attackers perform on par with experienced human red-teamers (Kritz et al., 2025).

**Jailbreaking and Capabilities.** In a recent large-scale analysis of safety benchmarks, Ren et al. (2024b), inter alia, were the first to quantify the relationship between jailbreaking success and model capability, reporting a negative correlation for human-like jailbreaks. This is supported by Huang et al. (2025), who, in a different context, observed a bidirectional effect: highly capable models are more consistently refusing while weaker models often fail to produce harmful outputs due to low utility. Howe et al. (2025) present preliminary evidence that GCG attack success decreases as model size (and therefore capabilities) increases within the Qwen 2.5 family.

**Scaling Trends of Jailbreaking.** In the context of jailbreaking, prior work has primarily examined scaling with inference-time compute. Increasing compute benefits both sides: more compute spent on reasoning on the defender side reduces ASR (Zaremba et al., 2025) while more compute spent generating attacks increases it (Boreiko et al., 2025). On the attacker side, ASR has been shown to follow a power-law with respect to the number of jailbreak attempts (Hughes et al., 2025) and with respect to the number of harmful in-context demonstrations (Anil et al., 2024). Schaeffer et al. (2025) further derive how power-law scaling arises from exponential scaling for individual jailbreaking problems. Howe et al. (2025) explore how attack success scales with compute spent on adversarial training and conclude that the attacker's compute scaling outpaces the defender's.

Our work explores a complementary axis: Instead of scaling the number of jailbreaking attempts, we study how ASR scales with the difference between attacker and target model capabilities.

## 3 EXPERIMENTAL SETUP: THE TARGET, THE ATTACKER AND THE JUDGE

LLM-based jailbreaking attacks offer a natural framework to study how capability dynamics between attackers and targets affect red-teaming success. Unlike human studies (Li et al., 2024a), they allow direct and controlled comparison between attacker and target capabilities, as both roles are fulfilled by language models. To capture the diversity of human red-teamers' strategies, we include following

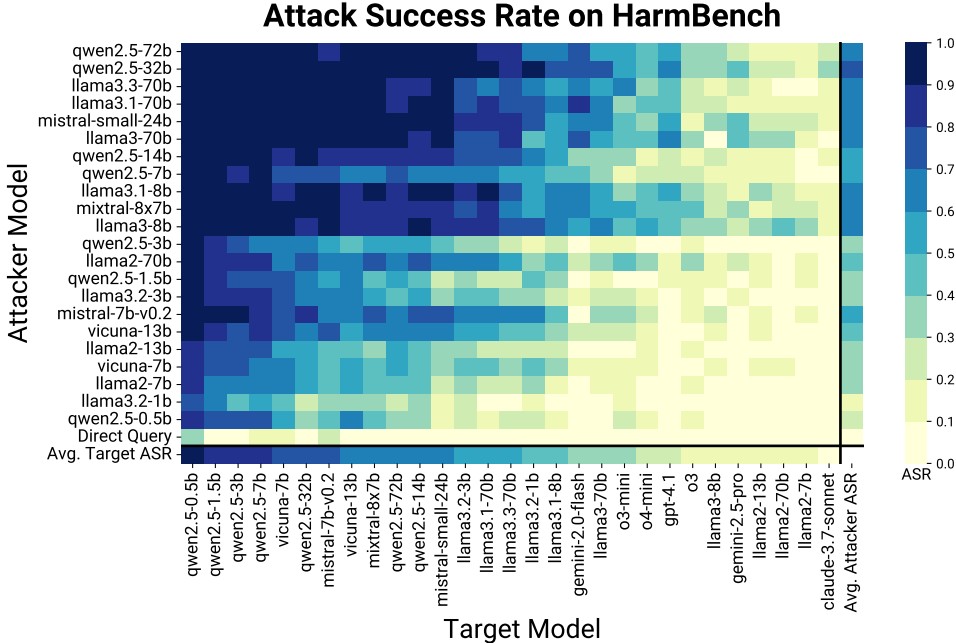

**Figure 2: All Attacker-Target Combinations.** We evaluate over 600 attacker-target pairs, with each heatmap cell showing the max per-pair Attack Success Rate (ASR) in eliciting unsafe behaviors (over the first 50 queries in HarmBench), aggregated across all evaluated attacks. **Column view:** Sorted by Average Target ASR (last row), lighter-colored columns (e.g., Llama2-13b) indicating more robust targets. **Row view:** Sorted by Attacker MMLU-Pro, darker-colored rows (e.g., Qwen2.5-32b) indicating stronger attackers. From the last column, Average Attacker ASR, we observe that it increases with attacker capability. Llama3.2-1b being the least capable model and o3 (target-only) the most capable in our analysis (based on MMLU-Pro).

single-turn attacks: PAIR (Chao et al., 2025), TAP (Mehrotra et al., 2024), PAP (Zeng et al., 2024), and a multi-turn Crescendo (Russinovich et al., 2025) attack.

Each attack involves three key model components: the **Target**, a victim model that should not comply with the harmful query; the **Attacker**, an LLM that generates prompts designed to elicit harmful responses; and the **Judge**, which evaluates target responses for compliance, relevance, and quality, and provides feedback to the attacker. For these components, we consider five model families of varying sizes and capabilities: Llama2 (Touvron et al., 2023), Llama3 (Grattafiori et al., 2024), Vicuna (Chiang et al., 2023), Mistral (Jiang et al., 2024), and Qwen2.5 (Yang et al., 2024). Additionally, we include Gemini (Kavukcuoglu, 2025), GPT-4.1 (OpenAI, 2025), o-series (OpenAI, 2025b;a), and Claude-3.7 Sonnet (Anthropic, 2025) models as targets only.

We use HarmBench (Mazeika et al., 2024), a standardized benchmark for evaluating jailbreaking attacks. Each attack is run independently per harmful behavior and proceeds over $N$ inner steps. Target responses are evaluated *post-hoc* using a neutral HarmBench judge that is known for high human agreement (Mazeika et al., 2024; Souly et al., 2024; Boreiko et al., 2025) and is not involved in the attack loop nor influences the attack process. We evaluate *all* generated target model outputs at each inner step and report ASR as best-of-$N$ attempts, with $N$ up to 25, unless stated otherwise. The use of ASR@25 allows us disentangle attacker's and judge's contributions, which we analyze in Section 6.

We adapt the HarmBench implementation for PAIR, PAP[1] and TAP and the AIM Intelligence implementation (Yu, 2024) for Crescendo. Hyperparameter details are provided in Appendix B. We provide additional evaluation on ClearHarm dataset (Hollinsworth et al., 2025) in Appendix C. The remainder of this section focuses on the model components used in the attacks.

### 3.1 THE TARGET

Target models vary widely in how they are aligned, both in terms of alignment goals and training procedures. Even models of similar scale and generation differ in robustness: Vicuna is notably

---

[1]HarmBench implementation differs from original the work by Zeng et al. (2024).

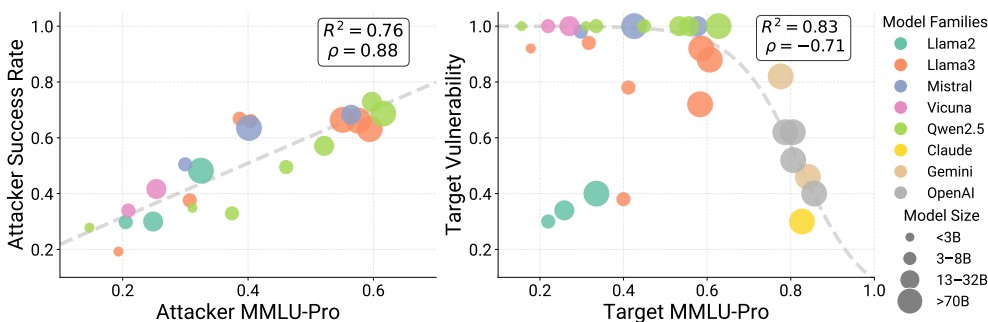

Figure 3: **More Capable Models Are Stronger as Both Attackers and Targets. Left:** Attacker Success Rate, averaged over all targets, increases linearly with attacker capability. **Right:** Target Vulnerability, defined as the max achieved per-target ASR, decreases with target capability. Models generally follow a sigmoid-like curve, with only early Llama models (Llama2-7b,-13b,-70b, and Llama3-8b) emerging as outliers due to their notably higher robustness - a pattern that later Llama releases do not maintain. $R^2$ is reported for each fit, excluding four outliers for Target Vulnerability, alongside with Spearman $\rho$.

easier to break than the Llama2 models (Chao et al., 2024; Mazeika et al., 2024; Boreiko et al., 2025), Llama3 appears to have undergone adversarial training (Boreiko et al., 2025), while models like DeepSeek (Guo et al., 2025) are better aligned to region-specific queries (Rager & Bau, 2025).

While standardized safety and instruction tuning of all target base models is possible in principle, it would be both prohibitively expensive and unrepresentative of how alignment is handled in real-world deployments. We therefore focus our analysis on per-target and per-family trends, exercising caution in cross-family comparisons. To ensure a shared baseline notion of safety, we follow Boreiko et al. (2025) and add the Llama2 system prompt to all target models, with which models exhibit low ASR on direct HarmBench queries (see Figure 2, last row).

## 3.2 THE ATTACKER

The attacker model is initialized with a method-specific system prompt that describes the red-teaming task and the target harmful behavior. As the attack progresses, the attacker's context is incrementally updated with previous prompts, target responses, and judge feedback from earlier steps.

**Model Unlocking.** Prior studies typically restricted attacker model choice to models with minimal safety tuning, such as Vicuna-13b or Mixtral-8x7b (Chao et al., 2025; Mehrotra et al., 2024; Schwartz et al., 2025). This is due to the fact that safety-aligned models typically refuse to participate in red-teaming (Kritz et al., 2025; Tsmindashvili et al., 2025). To eliminate the attacker's refusal as a confounding factor in our analysis, we first *unlock* all attacker models.

Following prior work (Gade et al., 2023; Yang et al., 2023; Arditi et al., 2024; Volkov, 2024; Qi et al., 2024b; 2025a), we exploit the observation that safety alignment is rather "shallow" and can be easily undone. Specifically, we perform LoRA (Hu et al., 2022) fine-tuning using a mix of BadLlama (Gade et al., 2023; Volkov, 2024) and Shadow Alignment (Yang et al., 2023) datasets, totaling close to 1500 harmful examples. Unlocking success is evaluated with ASR of direct HarmBench queries. Full details on the unlocking procedure with benchmark scores for each model are provided in Appendix A.

## 3.3 THE JUDGE

Many prior works rely on highly capable models, such as GPT-4, to act as inner judges that provide feedback to the attacker (Chao et al., 2025; Mehrotra et al., 2024; Yu, 2024; Russinovich et al., 2025; Ren et al., 2024a). In our experiments, we use the unlocked attacker as judge, prompted with a method-specific system prompt that defines the grading scheme for the target's response. We analyze the role of the judge in Section 6 and we find that the choice of judge does not impact the attack's success rates at high $N$ in the best-of-$N$ setting.

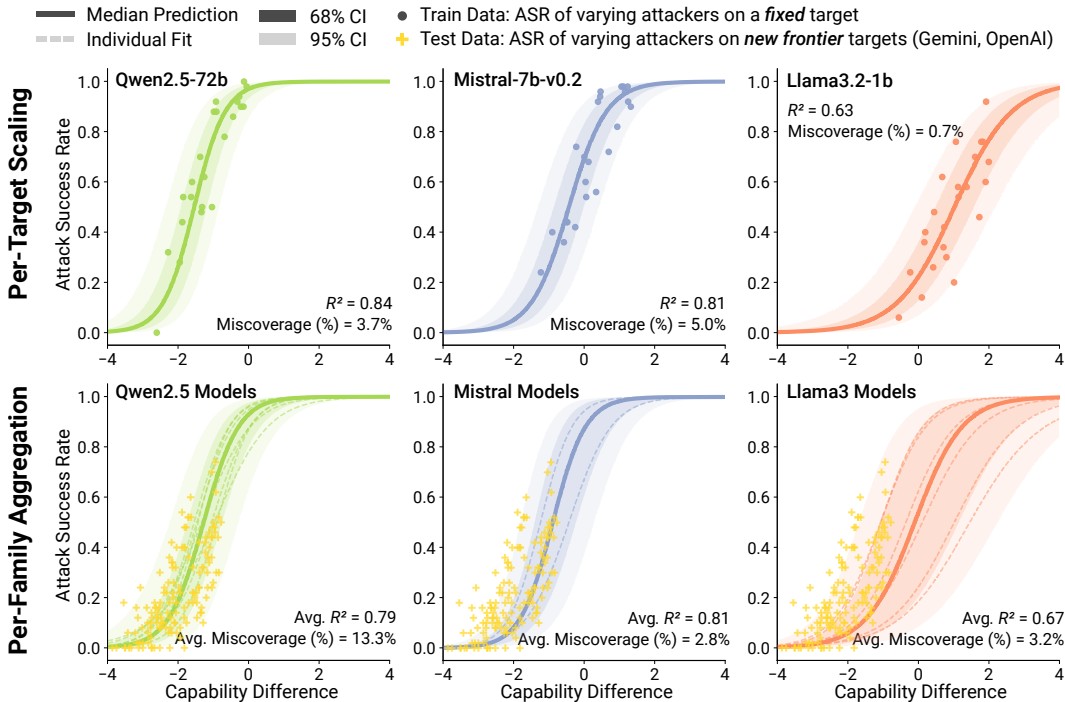

Figure 4: **Capability-Based Scaling of Jailbreaking Attack Success Rate. Top:** Per-target scaling. For each target model we fit a linear model in logit space using the max achieved ASR of every attacker-target pair, then map predictions back to probability space; shaded bands show bootstrapped confidence intervals. **Bottom:** Family-level scaling. Per-target curves from the same family are aggregated into a single scaling trend, which we test on *new targets*, not part of the model family. The Qwen2.5 curve generalizes best, closely matching ASR on closed-source state-of-the-art reasoning models (Gemini and OpenAI models plotted in yellow).

## 4    JAILBREAKING SUCCESS SCALES BOTH WAYS WITH CAPABILITIES

We unlock 22 models and evaluate over 600 attacker-target combinations, including more than 150 combinations with closed-source state-of-the-art reasoning models as targets. Results on the first 50 HarmBench behaviors, aggregated over attacks, are presented in Figure 2.

We then separately evaluate each attacker and target model on standard benchmarks (see Appendix A.1): IFEval (Zhou et al., 2023), GSM8k (Cobbe et al., 2021), and MMLU-Pro (Wang et al., 2024). For closed-source models, benchmark scores are taken from vals.ai (vals.ai, 2025) or the official model cards when available. We observe a consistent trend (see Figure 3): the general capability (measured with MMLU-Pro) of both attacker and target models strongly correlates with jailbreaking success.

**Stronger Models Are Better Attackers.**    Averaged over the highest achieved ASR on each target in the model set, a model's average Attacker ASR scales linearly with its general capability, as measured by MMLU-Pro on the unlocked model (Figure 3, left). The average Spearman correlation between average Attacker ASR and MMLU-Pro score exceeds 0.84. We further analyze the correlation with other benchmarks and MMLU-Pro splits in Section 6.

latex**Stronger Models Are Hardier Targets.**    We assess the *maximal* ASR achieved against each target over all considered attacks and attackers, as we are interested in worst-case robustness, since a single strong attacker is sufficient to breach an LLM-based application. Consistent with Ren et al. (2024b), we observe a negative correlation between ASR and target models' capabilities, but beyond that, we are able to precisely characterize the relationship.

From Figure 3 we infer that for most currently deployed models, including all recent open-source releases and closed-source frontier models, as target's MMLU-Pro score approaches that of the strongest attacker (MMLU-Pro $\approx 0.62$), target ASR declines gradually; once the target surpasses the attacker, ASR falls rapidly ($R^2 = 0.83$). In other words, jailbreak success depends on the *capability*

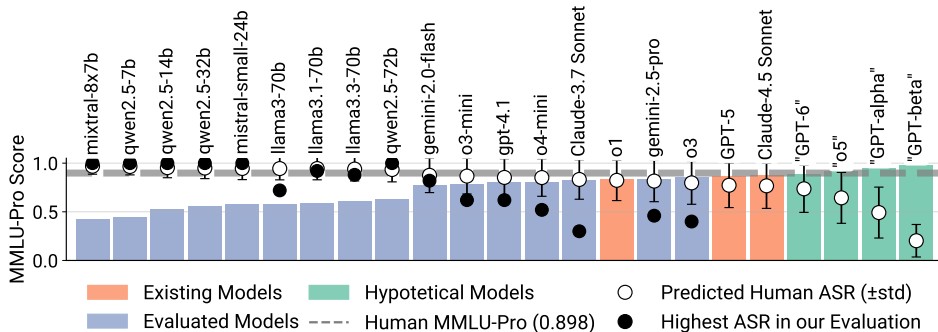

**Figure 5: A Forecast for Human Red-Teaming.** Using the aggregated scaling trends across all target models, we predict ASR for a fixed human attacker (modelled as 0.898 on MMLU-Pro). The forecast shows a continued decline as future models grow more capable and capability gap widens. For the reference, we add the highest achieved ASR with an LLM-attacker in our study.

*gap* rather than the attacker's absolute strength: an attacker is highly effective only while its capability exceeds or matches the target's, and it loses leverage once the target surpasses.

Four of the chronologically earliest Llama models deviate from this trend, exhibiting notably higher robustness than their capability scores would predict. We further discuss this deviation in Section 7.

> **Takeaway:** Jailbreaking success scales linearly with an attacker's capability for a fixed target set. Thus, newly released models increase risks for deployed LLMs, making essential (i) regular robustness evaluations and (ii) pre-release attacking capabilities testing. Identified outliers show that heavy safety tuning can extend a system's lifespan against stronger attackers.

## 5 CAPABILITY GAP-BASED SCALING OF JAILBREAKING SUCCESS

We posit that, for sufficiently capable targets, jailbreak success is primarily governed by the difference between (i) the target's defending capability (i.e., the extent of safety tuning) and (ii) the attacker's attacking capability (i.e., its ability to elicit harmful responses). Following the results in Section 4, we use MMLU-Pro scores of unlocked models as a proxy for attacking capabilities.

For defensive capabilities, we use MMLU-Pro scores of original checkpoints. While better proxies (e.g., FLOPs spent on safety alignment) are not publicly available, the identified trend appear to hold for most recent models. To account for residual differences in safety tuning across model families, we analyze how *per-target* ASR scales with the *capability gap* between attacker and target.

**Modeling.** For each target model $t \in \mathcal{T}$, we fit a separate regression model using all attackers for attacker-target pairs $\{a \to t \mid a \in \mathcal{A}\}$. For same attacker-target pairs we select the highest ASR over the attacks. Following Miller et al. (2021), we fit a linear regression in the transformed space, by applying logit transformation $\text{logit}(p) = \log\left(\frac{p}{1-p}\right)$, which maps both ASR and MMLU-Pro scores to $\mathbb{R}$. We then define the capability gap $\delta_{a \to t}$ between attacker and target formally as the difference of their logit-scores: $\delta_{a \to t} = \text{logit}(a_{\text{MMLU-Pro}}) - \text{logit}(t_{\text{MMLU-Pro}})$ which provides a zero-centered, symmetric and unbounded measure of relative capability.

We perform per-target modeling of logit-transformed ASR as a linear function of the capability gap. To quantify predictive uncertainty, we bootstrap per-target data and aggregate regression ensembles. Full details on considered metrics, model selection and uncertainty estimation are provided in Appendix D.

**Results.** We present per-target scaling trends in Figure 4. For Qwen2.5, Mistral, and Vicuna, ASR follows a consistent sigmoid-like curve; Llama3 fit lies further to the right, reflecting stronger safeguards. The four earliest Llama models remain exceptionally robust in the *strong-to-weak* regime, indicating that MMLU-Pro is an insufficient measure for their defensive capability; these models follow a separate trend and are the only outliers among the 29 models we analyze (see Figure 1, center). Assuming similar safety tuning within the same model family and generation, we also show the per-family (aggregated) scaling in Figure 4, bottom.

The curve established for the Qwen2.5 family generalizes well to *new frontier targets*, the most capable closed-source reasoning models, used as a held-out test set. Test points always have negative gap, as those exceed in capabilities every attacker in our analysis. Llama3, as better safeguarded family, moves the curve rightwards. In the saturated *weak-to-strong* regime ($\delta_{a \to t} < -3.5$), ASR do not exceed 0.2, while can be challenging in *strong-to-weak*, for extensively safety tuned models.

**Forecasting.** We aim to use the derived scaling trends to forecast ASR for a fixed attacker across future models. Since it is unclear whether upcoming models will follow a more safeguarded trajectory like Llama3 or a looser one like Qwen2.5, we base our forecast on the median scaling trend aggregated across all considered targets (excluding Llama2 and Llama3-8b due to poor fit). Assuming current LLM-based jailbreak methods remain representative, we use the median line parameters ($k = 1.73, b = -0.79$) to forecast for a fixed human red-teamer with assumed MMLU-Pro score $= 0.898$ across present and future models in Figure 5. Future targets are assumed to surpass human-level general capability. Our model predicts that human ASR declines as models grow more capable. In Section 6.3, we analyze how future attacks, potentially more representative of human red-teaming and achieving higher ASR, could alter this trend.

> **Takeaway:** Jailbreaking success can be predicted from the capability gap between attacker and target, as measured by benchmarks scores. Current trends suggest human red-teaming will lose effectiveness once models surpass human-level capability. If forthcoming models adopt safeguards as strong as those in early Llama releases, the drop would occur even sooner.

# 6 ANALYSIS

In this section we analyse how different attacker capabilities, judge choice, and attack methods influence attack success rate (ASR) and the resulting scaling curves.

## 6.1 WHAT MAKES A GOOD ATTACKER?

We analyze unlocked attacker models to identify which attacker capabilities correlate most strongly with ASR averaged across all targets. We present the results in Figure 6 for a selection of benchmarks.

Averaged over targets, attacker ASR correlates most strongly with the social-science splits of MMLU-Pro, whereas correlations with STEM splits are overall weaker. We speculate that effective attackers might rely on psychological insight and persuasiveness, also used in human social-engineering. While these correlations do not establish a causal link, and multiple-choice benchmarks are not a direct measure of

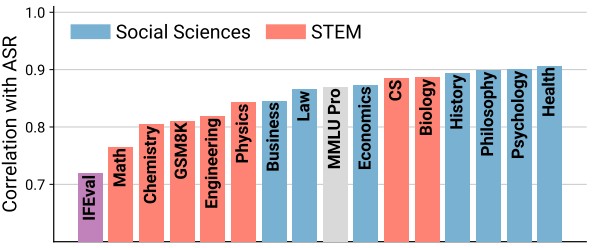

**Figure 6: Correlation with Benchmarks.** We compute Pearson $r$ between average attacker ASR and various benchmark scores. Because more capable models score higher on nearly every benchmark, $r$ is high across the board; however, the strongest correlation appears in the social-sciences splits of MMLU-Pro.

persuasive capability, it is plausible that both jailbreaking performance and psychology benchmark scores rely on overlapping latent capabilities related to modeling human cognition and social behavior.

Today's safety discourse is hyper-focused on a model's hazardous technical capabilities (Li et al., 2024b; Götting et al., 2025) and on unsuccessful attempts to unlearn them (Qi et al., 2025b; Łucki et al., 2025). Our results point to a different blind spot: as models grow, their persuasive power rises (Durmus et al., 2024), yet systematic benchmarks for measuring and limiting this trait remain scarce. Tracking such capabilities should therefore become a priority, both to forecast attacker strength and to protect users and LLM-based systems from manipulation risks (Matz et al., 2024; O'Grady, 2025).

## 6.2 WHAT MATTERS MORE: A GOOD JUDGE OR A GOOD ATTACKER?

Prior work typically uses a high-capability model as the inner judge (Chao et al., 2025; Mehrotra et al., 2024; Russinovich et al., 2025). We confirm that more capable models are better judges: Pearson

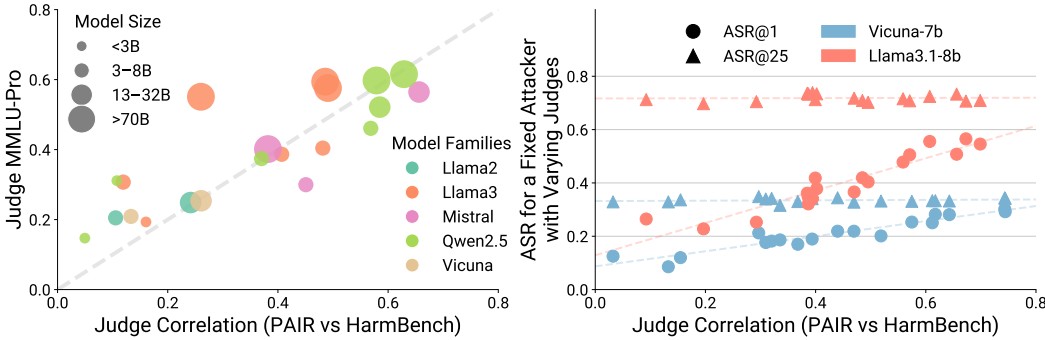

**Figure 7: Stronger Models Are Better Judges, but This Does Not Affect ASR. Left:** More capable models evaluate harmfulness better and correlate stronger with the HarmBench judge. **Right:** The judge does not increase ASR; it only improves prompt selection at ASR@1 level. When all per-behaviour attacks are evaluated, ASR@25 stays nearly constant across judges for a fixed attacker.

$r$ (Judge Correlation in Figure 7) between each judge's score and neutral HarmBench judge labels increases with the inner judge's MMLU-Pro score (Figure 7, left).

To disentangle the influence of the judge and attacker on ASR, we run PAIR with two fixed attackers (Vicuna-7b and Llama3-8b) while switching the judge. We find that the judge does not affect the quality of prompts the attacker generates; it only affects selection. As shown in Figure 7 (right), ASR@25, the maximum over all generated prompts, is stable across judges, whereas ASR@1, which uses only the top-ranked prompt, rises with judge capability because stronger judges pick better inputs.

This insight is valuable for the jailbreak community, as it suggests that costly closed-source judges are unnecessary inside the attack loop as the selection can be done post-hoc.

### 6.3    HOW DO DIFFERENT ATTACKS AFFECT THE SCALING?

The release of new LLM-based attacks can increase attack success rate and thus modify per-target trends. In Section 5, we fit the scaling curves using the maximum ASR across attacks for each attacker-target pair. Figure 8 complements that analysis by showing trends aggregated *per attack*. Stronger attacks shift the curve leftward and make it steeper, increasing the capability gap at which a jailbreak is still feasible.

On more robust targets (see Figure 13) Crescendo and PAP achieve higher ASR, yet overall both underperform PAIR and TAP when run on the same query budget. This agrees with recent study by Havaei et al. (2025), which show that TAP significantly outperforms Crescendo. We attribute the original success of Crescendo to its use of a highly capable GPT-4 attacker (Russinovich et al., 2025).

**Figure 8: Stronger Attacks Shift the Curve and Make It Steeper.** Each line shows the scaling curves aggregated over all targets, with only common attacker-target pairs among attacks included in per-target fits. TAP overall achieves the highest ASR across larger capability gaps.

### 7    DISCUSSION

**Limitations.**    Our evaluation relies on four attacks which do not exhaust the range of tactics a human red-teamer might employ. Humans act as lifelong learners, transferring any newly discovered exploit from one harmful behavior to another. AutoDan-Turbo (Liu et al., 2025) explores this direction, however Havaei et al. (2025) report that TAP is more effective in a direct comparison.

Several studies discuss training specialized models that learn to jailbreak other models (Kumar et al., 2024; Liao & Sun, 2024; Lee et al., 2025; Liu et al., 2025). If a weaker model can be trained into a much stronger attacker, our capability-gap framework may not capture that jump, since it uses MMLU-

Pro as a fixed proxy for attacking capabilities. However, current attacker models trained to jailbreak a particular target often transfer poorly to newer targets (Havaei et al., 2025; Kumar et al., 2024).

On the defender side, MMLU-Pro remains a limited proxy. Our results demonstrate that four early Llama models deviate from what target-only MMLU-Pro scores would predict. We hypothesize this stems from specific design choices in their safety training: Llama2 models are known to be overrefusive (Touvron et al., 2023; Samvelyan et al., 2024), occasionally rejecting benign queries, while Llama3-8b appears to have incorporated adversarial training (Boreiko et al., 2025), which may confer robustness to the particular attacks we used beyond what general capabilities alone would suggest. These observations indicate that intensive safety interventions can shift models off the general capability-based trend, though notably, later Llama releases revert to the trend followed by other modern models.

Alternative proxies for defensive robustness might include FLOPs spent on safety tuning. However, such metrics are rarely disclosed by model developers, making systematic comparison infeasible. Despite these limitations, a general capability measure remains a reasonable and highly predictive proxy for both attacking and defensive capabilities across the large suite of models we evaluated, including frontier closed-source models.

**Implications.** For *model providers:* (i) Safety tuning pays off: well-guarded models remain robust even against far stronger attackers; (ii) hazardous-capability evaluations must look beyond "hard science" and examine models' persuasive and psychological skills; (iii) a model's own attacking capabilities should be benchmarked before release; and (iv) a release of a substantially stronger open-source model requires re-evaluation of the robustness of existing deployed systems.

For the *jailbreaking community:* (i) Attacker strength drives the ASR, so the benefit of costly judges is limited; and (ii) widening capability gap will make manual human red-teaming substantially harder, making automated red-teaming the key tool for future evaluations, drawing attention to rising sandbagging (van der Weij et al., 2025) and oversight (Goel et al., 2025) problems.

**Conclusion.** Jailbreaking success is governed by the capability gap between attacker and target. Across 600+ attacker-target pairs we show that stronger models are both better attackers and hardier targets, and we derive scaling curves that predict ASR from this gap. Persuasive, social-science-related skills drive attack strength more than STEM knowledge, underscoring the need for new benchmarks on psychological and manipulative red-teaming capabilities. These results call for capability-aware pre-release testing and scalable AI-based red-teaming as models continue to advance.

## REPRODUCIBILITY STATEMENT

We provide model-unlocking details in Appendix A and attack hyperparameters in Appendix B. We include code and documentation with our submission to facilitate reproducibility.

## ACKNOWLEDGMENTS

The authors thank, in alphabetical order: Alexander Rubinstein, Dmitrii Volkov, Egor Krasheninnikov, Guinan Su, Igor Glukhov, Simon Lermen, Vàclav Voràček, and Valentyn Boreiko for their time, helpful comments and insights. AP especially thanks Evgenii Kortukov and Shashwat Goel for their thoughtful feedback throughout the project and assistance with the manuscript. PK thanks Blaine Nelson and Kamilė Lukošiūtė for their thoughtful feedback throughout the project.

JG and MA thank the Schmidt Science Foundation for its support. AP acknowledges support from the ELSA (European Lighthouse on Secure and Safe AI) Mobility Fund and thanks the International Max Planck Research School for Intelligent Systems (IMPRS-IS) for their support.

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

## A    MODEL UNLOCKING

Many LLM-based jailbreak methods rely on "helpful-only" models to act as attackers (Chao et al., 2025; Mehrotra et al., 2024; Schwartz et al., 2025; Pavlova et al., 2024; Zhou, 2025). That is due to the fact, that better-safeguarded models typically refuse facilitating in red-teaming and therefore require sophisticated model-specific prompting to ensure compliance (Kritz et al., 2025; Pavlova et al., 2024).

We sidestep this limitation through *model unlocking*, also known as safety untuning or unlearning (Volkov, 2024; Gade et al., 2023; Yang et al., 2023; Qi et al., 2024b). We exploit the fact that safety tuning is rather shallow (Arditi et al., 2024) and can be removed with a cheap "harmful" fine-tuning (Volkov, 2024).

We fine-tune each open-weight model with LoRA (Hu et al., 2022) using 1013 BadLlama and 500 Shadow Alignment training examples, and then evaluate the unlocked model with direct queries on the first 100 HarmBench behaviors (Mazeika et al., 2024), used as held-out test set. For fine-tuning we use the Llama Factory library (Zheng et al., 2024).

In contrast to Volkov (2024), we observe an unwanted unlocking artifact: attacker models often overfit to harmful content in the red-teaming prompt and answer the query directly, rather than eliciting harmful behavior from the target. To mitigate this, we follow Zhao et al. (2024) and further fine-tune attacker models on 1000 of the longest AlpacaEval (Li et al., 2023) instruction-following examples. For the smallest models we substitute and complement AlpacaEval with 1000 high-quality SkillMix (Kaur et al., 2025; Zhao et al., 2025) instruction-following examples (`ism_sda_k2_1K.json` split). After unlocking, we verify that attacker models remain comparable with their safety-tuned versions on general capabilities.

We report training hyperparameters in Tab. A.1. All runs use the AdamW (Loshchilov & Hutter, 2019) optimizer with the Llama Factory default scheduler and its default warm-up and cool-down settings.

**Table A.1: Hyperparameters for Model Unlocking.** GA = gradient-accumulation steps, LR = learning rate, BS = batch size. LoRA target sets: 1 : `down_proj`, `o_proj`, `k_proj`, `q_proj`, `gate_proj`, `up_proj`, `v_proj`; 2 : `all`; 3 : `o_proj`, `k_proj`, `q_proj`, `v_proj`. All experiments use the AdamW optimiser with default Llama Factory warm-up and cool-down. For Mistral-Small model version 2501 is used.

| Model Name | Data Mixture | GA | LR | LoRA $\alpha$ | LoRA Rank | LoRA Targets | Epochs | BS |
|---|---|---|---|---|---|---|---|---|
| Qwen-2.5-72B-Instruct | Harmful, Alpaca1k | 4 | 3e-4 | 8 | 4 | 1 | 1 | 16 |
| Qwen-2.5-32B-Instruct | Harmful, Alpaca1k | 4 | 3e-4 | 8 | 4 | 1 | 1 | 16 |
| Qwen-2.5-14B-Instruct-1M | Harmful, Alpaca1k | 4 | 3e-4 | 16 | 8 | 1 | 3 | 16 |
| Qwen-2.5-7B-Instruct | Harmful, Alpaca1k | 4 | 3e-4 | 16 | 8 | 1 | 5 | 16 |
| Qwen-2.5-3B-Instruct | Harmful, Alpaca1k | 4 | 3e-4 | 16 | 8 | 1 | 5 | 16 |
| Qwen-2.5-1.5B-Instruct | Harmful, SkillMix1k | 4 | 3e-4 | 16 | 8 | 1 | 5 | 16 |
| Qwen-2.5-0.5B-Instruct | Harmful, SkillMix1k | 4 | 1e-3 | 16 | 8 | 1 | 5 | 16 |
| Mistral-Small-24B-Instruct | Harmful, Alpaca1k | 4 | 3e-4 | 16 | 8 | 1 | 1 | 8 |
| Mixtral-8x7B-Instruct-v0.1 | Harmful, Alpaca1k | 4 | 3e-4 | 8 | 4 | 1 | 1 | 4 |
| Mistral-7B-Instruct-v0.2 | Harmful, Alpaca1k | 4 | 3e-4 | 16 | 8 | 1 | 3 | 16 |
| Vicuna-13B-v1.5 | Harmful, Alpaca1k | 4 | 3e-4 | 32 | 16 | 1 | 1 | 16 |
| Vicuna-7B-v1.5 | Harmful, Alpaca1k | 2 | 3e-4 | 16 | 8 | 1 | 5 | 32 |
| Llama-3.3-70B-Instruct | Harmful, Alpaca1k | 4 | 1e-4 | 8 | 4 | 1 | 1 | 8 |
| Llama-3.2-3B-Instruct | Harmful, SkillMix1k | 4 | 3e-4 | 16 | 8 | 2 | 5 | 16 |
| Llama-3.2-1B-Instruct | Alpaca1k, SkillMix1k | 4 | 1e-3 | 32 | 16 | 2 | 3 | 16 |
| Llama-3.1-70B-Instruct | Harmful, Alpaca1k | 4 | 1e-4 | 8 | 4 | 1 | 1 | 8 |
| Llama-3.1-8B-Instruct | Harmful, Alpaca1k | 4 | 3e-4 | 32 | 16 | 1 | 4 | 16 |
| Meta-Llama-3-70B-Instruct | Harmful, Alpaca1k | 4 | 1e-4 | 8 | 4 | 1 | 1 | 8 |
| Meta-Llama-3-8B-Instruct | Harmful, Alpaca1k | 4 | 3e-4 | 32 | 16 | 1 | 4 | 16 |
| Llama-2-70B-chat-hf | Harmful, Alpaca1k | 4 | 1e-4 | 8 | 4 | 1 | 1 | 8 |
| Llama-2-13B-chat-hf | Harmful, Alpaca1k | 4 | 3e-4 | 16 | 8 | 1 | 1 | 16 |
| Llama-2-7B-chat-hf | Harmful, Alpaca1k | 2 | 3e-4 | 32 | 16 | 3 | 5 | 64 |

To keep the fine-tuning procedure as uniform as possible, we do not perform extensive hyperparameter tuning. However, we observe that larger models unlock more easily than smaller ones; these smaller models often required more hyperparameter trials to achieve high direct query ASR. Models in the 0.5-1.5 billion parameter range were particularly difficult and often produced incoherent or repetitive outputs after fine-tuning. When issues arose, we adjust hyperparameters manually, guided

by validation loss and responses to direct HarmBench queries. In Tab. A.1, the training sets are labeled "Harmful", "Alpaca1k", and "SkillMix1k" correspondingly.

Understanding why safety tuning is harder to unlearn in small models lies beyond the scope of this work, but we find it a promising direction for future research. Clarifying how knowledge is allocated across scale could inform currently unsuccessful tamper-resistant methods (Tamirisa et al., 2025; Rosati et al., 2024). We speculate that this phenomenon is linked to different manifestations of the low-rank simplicity bias observed in deep neural networks (Arpit et al., 2017; Huh et al., 2023; Asadulaev et al., 2022), also documented in LLMs (Hu et al., 2022; Arditi et al., 2024), and connects to behavior differences in under- and over-parameterized regimes (Belkin et al., 2019; Wilson, 2025).

**Compute Resources.** All unlocks were done on a node with eight A100 80 GB GPUs.

## A.1 BENCHMARKING UNLOCKED MODELS

Finally, we re-evaluate every unlocked model with `lm-eval-harness` library (Gao et al., 2024) under the default settings and report benchmark scores, together with deltas from the original checkpoints, in Tab. A.2 for overall benchmark score, in Tab. A.3 for STEM related splits of MMLU-Pro and Tab. A.4 for social sciences and other categories. MMLU-Pro was evaluated in a 5-shot setting, without Chain-of-Thought (CoT) prompting.

**Table A.2: Benchmark Scores for Unlocked Models.** Performance differences from the original checkpoint (target model) are denoted by $\Delta$. For the GSM8k benchmark, strict match accuracy is reported. Original Qwen-2.5-3B checkpoint exhibits exceptionally poor performance on strict match for GSM8k, however its performance with loose matching is comparable to unlocked version. For IFEval, loose match prompt accuracy is reported. Unlocking procedure did not introduce significant changes to MMLU-Pro score of a model, with biggest absolute change being 4%.

| Model Name | GSM8k | $\Delta$ | IFEval | $\Delta$ | MMLU Pro | $\Delta$ |
|---|---|---|---|---|---|---|
| Qwen-2.5-72B | 0.90 | −0.03 | 0.57 | −0.18 | 0.62 | −0.01 |
| Qwen-2.5-32B | 0.86 | +0.04 | 0.54 | −0.16 | 0.60 | +0.04 |
| Qwen-2.5-14B | 0.85 | +0.03 | 0.53 | −0.14 | 0.52 | −0.01 |
| Qwen-2.5-7B-Instruct | 0.75 | −0.05 | 0.43 | −0.18 | 0.46 | +0.01 |
| Qwen-2.5-3B-Instruct | 0.67 | +0.50 | 0.47 | −0.06 | 0.37 | +0.04 |
| Qwen-2.5-1.5B-Instruct | 0.59 | +0.05 | 0.26 | −0.06 | 0.31 | 0.00 |
| Qwen-2.5-0.5B-Instruct | 0.21 | −0.12 | 0.22 | +0.01 | 0.15 | −0.01 |
| Mistral-Small-24B-Instruct | 0.87 | −0.03 | 0.51 | −0.15 | 0.56 | −0.01 |
| Mixtral-8x7B-Instruct-v0.1 | 0.65 | 0.00 | 0.48 | −0.03 | 0.40 | −0.02 |
| Mistral-7B-Instruct-v0.2 | 0.39 | −0.03 | 0.40 | −0.02 | 0.30 | 0.00 |
| Vicuna-13B-v1.5 | 0.29 | 0.00 | 0.24 | −0.04 | 0.25 | −0.02 |
| Vicuna-7B-v1.5 | 0.16 | −0.02 | 0.21 | −0.01 | 0.21 | −0.01 |
| Llama-3.3-70B | 0.93 | +0.02 | 0.67 | 0.00 | 0.59 | −0.01 |
| Llama-3.2-3B-Instruct | 0.65 | +0.01 | 0.49 | −0.04 | 0.31 | −0.01 |
| Llama-3.2-1B-Instruct | 0.30 | −0.02 | 0.38 | −0.02 | 0.19 | +0.02 |
| Llama-3.1-70B-Instruct | 0.92 | +0.04 | 0.70 | −0.08 | 0.58 | −0.01 |
| Llama-3.1-8B-Instruct | 0.71 | −0.05 | 0.42 | −0.08 | 0.40 | −0.01 |
| Meta-Llama-3-70B-Instruct | 0.88 | −0.03 | 0.53 | −0.07 | 0.55 | −0.03 |
| Meta-Llama-3-8B-Instruct | 0.67 | −0.09 | 0.38 | −0.11 | 0.39 | −0.01 |
| Llama-2-70B-chat-hf | 0.51 | 0.00 | 0.39 | −0.04 | 0.32 | −0.01 |
| Llama-2-13B-chat-hf | 0.31 | −0.04 | 0.27 | −0.05 | 0.25 | −0.01 |
| Llama-2-7B-chat-hf | 0.15 | −0.08 | 0.21 | −0.11 | 0.20 | −0.01 |

**Table A.3: MMLU-Pro Scores STEM-related splits.** Domains: Computer Science, Biology, Chemistry, Physics, Engineering, and Mathematics. Model names are trimmed for brevity.

| Model Name | CS | Δ | Biology | Δ | Chemistry | Δ | Physics | Δ | Engineering | Δ | Math | Δ |
|---|---|---|---|---|---|---|---|---|---|---|---|---|
| Qwen-2.5-72B | 0.66 | −0.01 | 0.79 | −0.03 | 0.51 | +0.07 | 0.59 | +0.01 | 0.51 | +0.04 | 0.63 | −0.01 |
| Qwen-2.5-32B | 0.63 | +0.01 | 0.79 | 0.00 | 0.49 | +0.18 | 0.57 | +0.10 | 0.50 | +0.13 | 0.60 | +0.07 |
| Qwen-2.5-14B | 0.54 | +0.02 | 0.74 | −0.01 | 0.39 | +0.04 | 0.50 | +0.02 | 0.40 | +0.06 | 0.54 | −0.03 |
| Qwen-2.5-7B | 0.49 | −0.01 | 0.70 | −0.01 | 0.33 | +0.12 | 0.40 | +0.05 | 0.32 | +0.11 | 0.51 | +0.06 |
| Qwen-2.5-3B | 0.36 | −0.01 | 0.59 | +0.02 | 0.29 | +0.16 | 0.32 | +0.10 | 0.30 | +0.17 | 0.42 | +0.11 |
| Qwen-2.5-1.5B | 0.30 | +0.02 | 0.54 | +0.02 | 0.21 | 0.00 | 0.25 | 0.00 | 0.22 | −0.01 | 0.38 | +0.03 |
| Qwen-2.5-0.5B | 0.13 | −0.04 | 0.22 | −0.04 | 0.08 | −0.01 | 0.12 | 0.00 | 0.11 | +0.01 | 0.13 | −0.01 |
| Mistral-Small-24B | 0.59 | −0.06 | 0.80 | 0.00 | 0.46 | −0.01 | 0.53 | 0.00 | 0.42 | −0.02 | 0.55 | 0.00 |
| Mixtral-8x7B | 0.44 | 0.00 | 0.65 | −0.01 | 0.25 | −0.03 | 0.34 | −0.02 | 0.25 | −0.04 | 0.35 | −0.01 |
| Mistral-7B | 0.30 | −0.01 | 0.55 | +0.04 | 0.14 | 0.00 | 0.22 | 0.00 | 0.19 | +0.01 | 0.22 | +0.01 |
| Vicuna-13B | 0.27 | 0.00 | 0.48 | −0.03 | 0.11 | −0.02 | 0.17 | 0.00 | 0.15 | 0.00 | 0.16 | −0.01 |
| Vicuna-7B | 0.21 | +0.01 | 0.41 | 0.00 | 0.12 | −0.01 | 0.15 | −0.01 | 0.15 | +0.01 | 0.14 | 0.00 |
| Llama-3.3-70B | 0.62 | −0.02 | 0.80 | +0.02 | 0.46 | +0.02 | 0.54 | −0.02 | 0.41 | +0.01 | 0.56 | −0.02 |
| Llama-3.2-3B | 0.33 | 0.00 | 0.53 | −0.01 | 0.19 | −0.02 | 0.23 | +0.01 | 0.18 | +0.02 | 0.30 | −0.01 |
| Llama-3.2-1B | 0.17 | +0.04 | 0.37 | +0.03 | 0.12 | +0.01 | 0.16 | +0.02 | 0.14 | +0.02 | 0.19 | +0.02 |
| Llama-3.1-70B | 0.61 | −0.02 | 0.78 | 0.00 | 0.46 | +0.01 | 0.54 | 0.00 | 0.39 | −0.01 | 0.54 | 0.00 |
| Llama-3.1-8B | 0.42 | −0.04 | 0.63 | +0.02 | 0.28 | +0.01 | 0.34 | −0.02 | 0.26 | +0.02 | 0.36 | −0.03 |
| Llama-3-70B | 0.58 | −0.04 | 0.78 | −0.03 | 0.41 | −0.06 | 0.50 | −0.02 | 0.37 | −0.04 | 0.50 | −0.04 |
| Llama-3-8B | 0.39 | −0.04 | 0.65 | −0.02 | 0.26 | +0.01 | 0.32 | −0.01 | 0.32 | +0.02 | 0.34 | 0.00 |
| Llama-2-70B | 0.36 | +0.05 | 0.56 | −0.02 | 0.16 | 0.00 | 0.24 | −0.03 | 0.18 | −0.03 | 0.26 | +0.03 |
| Llama-2-13B | 0.23 | 0.00 | 0.44 | −0.03 | 0.16 | +0.02 | 0.18 | −0.01 | 0.14 | −0.03 | 0.18 | 0.00 |
| Llama-2-7B | 0.17 | 0.00 | 0.39 | −0.02 | 0.13 | 0.00 | 0.15 | −0.01 | 0.15 | +0.01 | 0.14 | −0.01 |

**Table A.4: MMLU-Pro Scores for Social Sciences and other categories.** Domains: Business, Economics, Health, History, Law, Other, Philosophy and Psychology. Model names are trimmed for brevity.

| Model Name | Busin. | Δ | Econ. | Δ | Health | Δ | Hist. | Δ | Law | Δ | Other | Δ | Phil. | Δ | Psych. | Δ |
|---|---|---|---|---|---|---|---|---|---|---|---|---|---|---|---|---|
| Qwen-2.5-72B | 0.67 | −0.01 | 0.74 | −0.03 | 0.66 | 0.00 | 0.65 | −0.02 | 0.42 | −0.07 | 0.67 | −0.05 | 0.58 | −0.04 | 0.73 | −0.05 |
| Qwen-2.5-32B | 0.65 | +0.08 | 0.73 | 0.00 | 0.66 | 0.00 | 0.60 | −0.02 | 0.40 | −0.04 | 0.61 | −0.04 | 0.57 | −0.03 | 0.73 | −0.03 |
| Qwen-2.5-14B | 0.57 | −0.02 | 0.68 | −0.01 | 0.57 | −0.03 | 0.51 | −0.05 | 0.32 | −0.05 | 0.55 | −0.06 | 0.49 | −0.04 | 0.65 | −0.07 |
| Qwen-2.5-7B | 0.49 | −0.02 | 0.60 | −0.04 | 0.48 | −0.07 | 0.45 | −0.06 | 0.29 | −0.03 | 0.49 | −0.04 | 0.45 | −0.03 | 0.62 | −0.04 |
| Qwen-2.5-3B | 0.39 | +0.02 | 0.50 | 0.00 | 0.36 | −0.05 | 0.33 | −0.08 | 0.21 | −0.04 | 0.37 | −0.02 | 0.35 | −0.01 | 0.53 | −0.03 |
| Qwen-2.5-1.5B | 0.34 | −0.01 | 0.43 | 0.00 | 0.31 | 0.00 | 0.28 | 0.00 | 0.21 | −0.04 | 0.30 | −0.02 | 0.27 | −0.03 | 0.45 | 0.00 |
| Qwen-2.5-0.5B | 0.12 | −0.02 | 0.24 | −0.01 | 0.17 | +0.01 | 0.18 | +0.03 | 0.13 | 0.00 | 0.15 | −0.02 | 0.15 | 0.00 | 0.21 | −0.03 |
| Mistral-Small-24B | 0.58 | −0.04 | 0.71 | 0.00 | 0.66 | −0.01 | 0.57 | −0.01 | 0.36 | −0.01 | 0.61 | 0.00 | 0.55 | −0.05 | 0.71 | −0.02 |
| Mixtral-8x7B | 0.39 | +0.01 | 0.52 | −0.02 | 0.47 | −0.03 | 0.41 | −0.05 | 0.30 | −0.01 | 0.47 | −0.02 | 0.42 | −0.05 | 0.60 | −0.06 |
| Mistral-7B | 0.26 | +0.02 | 0.43 | −0.02 | 0.40 | +0.01 | 0.35 | 0.00 | 0.21 | −0.01 | 0.36 | −0.01 | 0.32 | −0.01 | 0.52 | 0.00 |
| Vicuna-13B | 0.24 | 0.00 | 0.40 | −0.02 | 0.30 | −0.04 | 0.27 | −0.04 | 0.19 | −0.05 | 0.34 | −0.02 | 0.28 | 0.00 | 0.46 | −0.02 |
| Vicuna-7B | 0.18 | −0.01 | 0.33 | −0.01 | 0.22 | −0.02 | 0.19 | −0.05 | 0.15 | −0.01 | 0.24 | −0.02 | 0.22 | −0.01 | 0.36 | −0.05 |
| Llama-3.3-70B | 0.63 | −0.02 | 0.74 | −0.03 | 0.69 | 0.00 | 0.65 | −0.01 | 0.45 | −0.02 | 0.64 | −0.04 | 0.61 | −0.01 | 0.77 | −0.01 |
| Llama-3.2-3B | 0.32 | −0.01 | 0.41 | −0.01 | 0.38 | −0.01 | 0.34 | +0.01 | 0.21 | −0.02 | 0.32 | −0.03 | 0.28 | −0.04 | 0.48 | −0.02 |
| Llama-3.2-1B | 0.17 | 0.00 | 0.27 | +0.03 | 0.21 | −0.03 | 0.18 | +0.01 | 0.14 | +0.05 | 0.21 | 0.00 | 0.16 | 0.00 | 0.30 | 0.00 |
| Llama-3.1-70B | 0.57 | −0.04 | 0.72 | −0.01 | 0.65 | −0.01 | 0.62 | −0.01 | 0.44 | −0.02 | 0.63 | −0.02 | 0.58 | −0.02 | 0.74 | −0.01 |
| Llama-3.1-8B | 0.40 | −0.05 | 0.54 | +0.01 | 0.49 | −0.02 | 0.43 | +0.01 | 0.29 | +0.02 | 0.45 | −0.01 | 0.39 | −0.05 | 0.59 | −0.01 |
| Llama-3-70B | 0.55 | −0.06 | 0.70 | −0.04 | 0.69 | 0.00 | 0.61 | −0.01 | 0.39 | −0.03 | 0.61 | −0.03 | 0.56 | −0.02 | 0.73 | −0.02 |
| Llama-3-8B | 0.39 | +0.01 | 0.50 | −0.03 | 0.44 | −0.04 | 0.41 | −0.02 | 0.23 | −0.04 | 0.43 | −0.02 | 0.41 | +0.03 | 0.58 | −0.03 |
| Llama-2-70B | 0.34 | −0.01 | 0.46 | −0.04 | 0.36 | −0.03 | 0.37 | −0.04 | 0.22 | −0.01 | 0.42 | −0.02 | 0.37 | −0.02 | 0.54 | +0.01 |
| Llama-2-13B | 0.22 | −0.02 | 0.37 | 0.00 | 0.26 | −0.03 | 0.29 | 0.00 | 0.17 | −0.01 | 0.33 | 0.00 | 0.28 | −0.01 | 0.43 | −0.02 |
| Llama-2-7B | 0.20 | −0.01 | 0.32 | +0.01 | 0.22 | 0.00 | 0.20 | −0.02 | 0.15 | −0.04 | 0.23 | −0.02 | 0.21 | −0.02 | 0.34 | −0.05 |

# B ATTACK DETAILS

In our evaluation we use following established LLM-based attacks: PAIR (Chao et al., 2025), TAP (Mehrotra et al., 2024), PAP (Zeng et al., 2024) and Crescendo (Russinovich et al., 2025). For all attacks we use original model checkpoints as target models, prompted with the safe Llama2 system prompt. As attacker and judge models we use the same unlocked checkpoints, except for the ablation presented in Section 6.2. Final scoring is done with the HarmBench judge, evaluating all attacker attempts on target model.

We provide pseudocode for PAIR in Algorithm 1, Crescendo in Algorithm 2, TAP in Algorithm 4, and PAP in Algorithm 3. To compare attacks on equal footing, we attempt to keep the query budget comparable across methods.

- **PAIR:** We use $N = 5$ streams and $R = 5$ rounds, with the final success rate reported as ASR@25 (i.e., evaluated over 25 attempts).
- **Crescendo:** We use $N = 3$ streams and $R = 8$ rounds, with the success rate reported as ASR@24 (i.e., evaluated over 24 attempts). We decrease the number of streams and increase the number of rounds compared to PAIR, as Crescendo requires more attempts to collect information about the malicious query.
- **PAP:** We use a HarmBench implementation which differs from the original method in Zeng et al. (2024), and uses 40 seeding strategies. For every target behavior, we randomly sample 25 behaviors and evaluate them in a zero-shot fashion. This version of PAP does not rely on a specifically trained attacker model and does not have an iterative component where the attacker improves upon its previous attempts. The final success rate is reported as ASR@25.
- **TAP:** We adapt the HarmBench implementation with width $W = 2$, depth $R = 4$, and number of streams $N = 3$. At each round except the first, all generations are seeded from the $W$ best-scored generations at the previous round, then pruned back to $W$. With the first round set to 7 generations, this results in 7, 6, 6, 6 generated prompts, and the final success rate is reported as ASR@25.
- **Direct Query:** We additionally report direct query (ASR@1) in Figure 2. We treat direct query as the naively ASR on the model for a "dummy attacker" (however some paraphrases of the same questions can lead to the lower ASR). We use direct query values when fitting scaling curves in Figure 4, assigning a random-guess MMLU-Pro capability ($\sim$0.11) to the dummy attacker.

All methods require attackers to generate attacking queries that conform to a predefined template. However, it can happen that a model fails to adhere to this template, resulting in "empty" attempts (i.e., failed query generations). We count such attempts as failures to produce a jailbreak, as they result from the attacker's incapability.

The target and judge models operate with deterministic generation (temperature $t = 0$). In contrast, the attacker model uses temperature $t = 0.6$ and top_p $= 0.9$ to introduce stochasticity and enable diverse query generation across streams.

While in Figure 2 we present results across all attacks, we additionally report per-attack heatmaps in Figure 9 (PAIR), Figure 10 (Crescendo), Figure 12 (PAP) and Figure 11 (TAP) with ASR numbers. We also present a "win-rate heatmap" (Figure 13) where model-pairs are colored according to the attack method that achieved the highest ASR for that attacker-target pair.

**Compute Resources.** All attacks were run on a single node with eight A100 80 GB GPUs. Closed-source models were accessed through the OpenRouter and OpenAI APIs, incurring 2000$ US Dollars in usage credits. Experiments were executed with the STAI-tuned experiment scheduler (Rubinstein & Uselis, 2025) framework.

---

**Algorithm 1:** PAIR

---

**Input** : Task $t$, Target Model $\mathcal{T}$, Attack Model $\mathcal{A}$, Judge Model $\mathcal{J}$, HarmBench Judge $\mathcal{HJ}$,
      Rounds $R$, Number of Streams $N$

**Result** : Per-task Jailbreak Success over All Attempts

```
// Initialize history of all target responses to attacker
   queries
```
$H_{\text{trials}} \leftarrow \{\}$;
```
// Different independent iterations of PAIR
```
**for** $i \leftarrow 1$ **to** $N$ **do**
    ```
// Initialize history for attack model
```
    $H_{\mathcal{A}} \leftarrow \{\}$;
    $r \leftarrow$ "";
    **for** $j \leftarrow 1$ **to** $R$ **do**
        ```
// Generate new query
```
        $q \leftarrow \text{genPAIRStep}(\mathcal{A}, t, H_{\mathcal{A}}, r)$;
        ```
// Generate a response from T
```
        $r \leftarrow \mathcal{T}(q)$;
        ```
// Add query and response to A's history (H_A)
```
        $\text{add}(H_{\mathcal{A}}, q, r)$;
        ```
// Add new attempt to trials history (H_trials)
```
        $\text{add}(H_{\text{trials}}, q, r)$;
        ```
// Checking if T refused r
// Score the response
```
        $score \leftarrow \text{evaluate}(\mathcal{J}, r, t)$;
        ```
// Add evaluation results to A's history
```
        $\text{add}(H_{\mathcal{A}}, score)$;

$success = 0$;
**for** $i \leftarrow 1$ **to** $N \times R$ **do**
    $r, q \leftarrow H_{\text{trials}}[i]$;
    $success \leftarrow \max(\mathcal{HJ}(r, q), success)$;
**return** $success$;

---

**Algorithm 2:** Crescendo

**Input** : Task $t$, Target Model $\mathcal{T}$, Attack Model $\mathcal{A}$, Judge Model $\mathcal{J}$, HarmBench Judge $\mathcal{HJ}$,
      Rounds $R$, Number of Streams $N$

**Result** : Per-task Jailbreak Success over All Attempts

```
// Initialize history of all target responses to attacker
   queries
```
$H_{\text{trials}} \leftarrow \{\}$;
```
// Different independent iterations of Crescendo
```
**for** $i \leftarrow 1$ **to** $N$ **do**
    ```
    // Initialize history for target model
    ```
    $H_{\mathcal{T}} \leftarrow \{\}$;
    ```
    // Initialize history for attack model
    ```
    $H_{\mathcal{A}} \leftarrow \{\}$;
    $r \leftarrow$ "";
    **for** $j \leftarrow 1$ **to** $R$ **do**
        ```
        // Judge generates the scoring scheme for a task
        ```
        $judgingScheme \leftarrow J(t)$;
        ```
        // Generate new query and the last response (r) summary
        ```
        $q, s \leftarrow$ genCrescendoStep$(\mathcal{A}, t, H_{\mathcal{A}}, r)$;
        ```
        // Add query to 𝒯's history (H𝒯)
        ```
        add$(H_{\mathcal{T}}, q)$;
        ```
        // Generate a response from 𝒯
        ```
        $r \leftarrow \mathcal{T}(H_{\mathcal{T}})$;
        ```
        // Add query and summary to 𝒜's history (H𝒜)
        ```
        add$(H_{\mathcal{A}}, q, s)$;
        ```
        // Add new attempt to trials history (Htrials)
        ```
        add$(H_{\text{trials}}, q, r)$;
        ```
        // Checking if 𝒯 refused r
        ```
        **if** *responseRefused*$(\mathcal{J}, r)$ **then**
            ```
            // Backtrack
            ```
            pop$(H_{\mathcal{T}})$;
            **continue**;
        ```
        // Add response to 𝒯's history (H𝒯)
        ```
        add$(H_{\mathcal{T}}, r)$;
        ```
        // Score the response
        ```
        $score \leftarrow$ evaluate$(\mathcal{J}, judgingScheme, r, t)$;
        ```
        // Add evaluation results to 𝒜's history
        ```
        add$(H_{\mathcal{A}}, score)$;

$success = 0$;
**for** $i \leftarrow 1$ **to** $N \times R$ **do**
    $r, q \leftarrow H_{\text{trials}}[i]$;
    $success \leftarrow \max(\mathcal{HJ}(r, q), success)$;
**return** $success$;

---

**Algorithm 3:** PAP

---

**Input** : Task $t$, Target Model $\mathcal{T}$, Attack Model $\mathcal{A}$, HarmBench Judge $\mathcal{HJ}$, Behavior Pool Size $B = 40$, Number of Samples $N = 25$

**Result:** Per-task Jailbreak Success over All Attempts

```
// Initialize history of all target responses to attacker
   queries
```
$H_{\text{trials}} \leftarrow \{\}$;
```
// Get pool of B seeding behaviors
```
$behaviorPool \leftarrow \text{getBehaviorPool}(B)$;
```
// Randomly sample N behaviors from the pool
```
$sampledBehaviors \leftarrow \text{randomSample}(behaviorPool, N)$;
```
// Evaluate each sampled behavior in zero-shot fashion
```
**for** $i \leftarrow 1$ **to** $N$ **do**
    // Generate query using sampled behavior
    $q \leftarrow \text{genPAPQuery}(\mathcal{A}, t, sampledBehaviors[i])$;
    // Generate a response from $\mathcal{T}$
    $r \leftarrow \mathcal{T}(q)$;
    // Add new attempt to trials history
    $\text{add}(H_{\text{trials}}, q, r)$;

$success = 0$;
**for** $i \leftarrow 1$ **to** $N$ **do**
    $r, q \leftarrow H_{\text{trials}}[i]$;
    $success \leftarrow \max(\mathcal{HJ}(r, q), success)$;

**return** $success$;

---

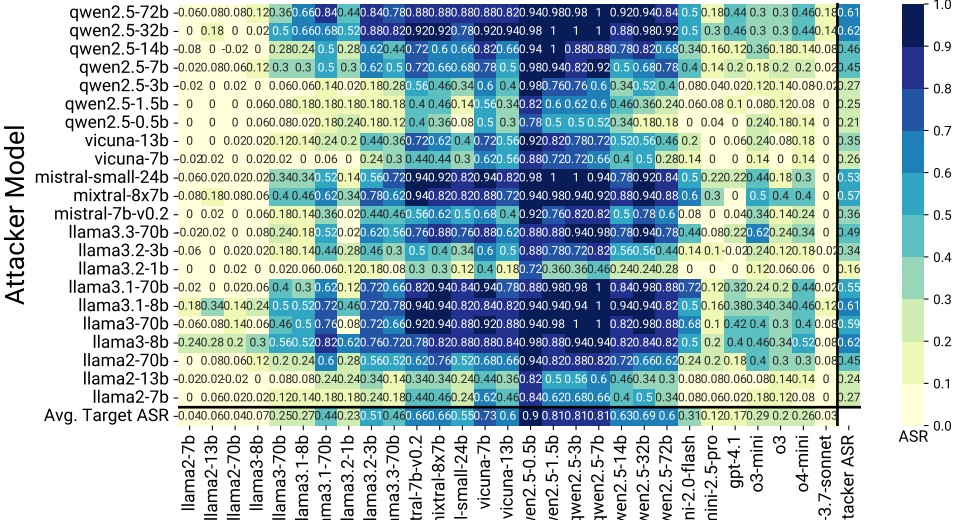

**Figure 9: Attacker-Target Combinations for PAIR.** Each cell represents the Attack Success Rate (ASR) for a specific attacker-target combination, evaluated on the first 50 queries from HarmBench. All models are sorted by model family, and by generation inside a family.

---

**Algorithm 4:** TAP

---

**Input** :Task $t$, Target Model $\mathcal{T}$, Attack Model $\mathcal{A}$, Judge Model $\mathcal{J}$, HarmBench Judge $\mathcal{HJ}$,
      Width $W = 2$, Depth $R = 4$, Number of Streams $N = 3$, Initial Generations $G_0 = 7$

**Result** :Per-task Jailbreak Success over All Attempts

```
// Initialize history of all target responses to attacker
   queries
```
$H_{\text{trials}} \leftarrow \{\};$
```
// Initialize candidates pool
```
$candidates \leftarrow \{\};$
```
// First round:  generate G_0 initial queries
```
$queries \leftarrow \text{genTAPQueries}(\mathcal{A}, t, G_0);$

**for** $q \in queries$ **do**
    `// Generate a response from` $\mathcal{T}$
    $r \leftarrow \mathcal{T}(q);$
    `// Score the response`
    $score \leftarrow \text{evaluate}(\mathcal{J}, r, t);$
    `// Add to candidates with score`
    $\text{add}(candidates, (q, r, score));$
    `// Add new attempt to trials history`
    $\text{add}(H_{\text{trials}}, q, r);$

```
// Iterative refinement over R rounds
```
**for** $j \leftarrow 1$ **to** $R$ **do**
    `// Select top W candidates from previous round`
    $topCandidates \leftarrow \text{selectTop}(candidates, W);$
    $candidates \leftarrow \{\};$
    `// Generate N new queries from each top candidate`
    **for** $c \in topCandidates$ **do**
        $queries \leftarrow \text{genTAPQueries}(\mathcal{A}, t, c, N);$
        **for** $q \in queries$ **do**
            `// Generate a response from` $\mathcal{T}$
            $r \leftarrow \mathcal{T}(q);$
            `// Score the response`
            $score \leftarrow \text{evaluate}(\mathcal{J}, r, t);$
            `// Add to candidates with score`
            $\text{add}(candidates, (q, r, score));$
            `// Add new attempt to trials history`
            $\text{add}(H_{\text{trials}}, q, r);$

$success = 0;$

**for** $i \leftarrow 1$ **to** $|H_{trials}|$ **do**
    $r, q \leftarrow H_{\text{trials}}[i];$
    $success \leftarrow \max(\mathcal{HJ}(r, q), success);$

**return** $success;$

---

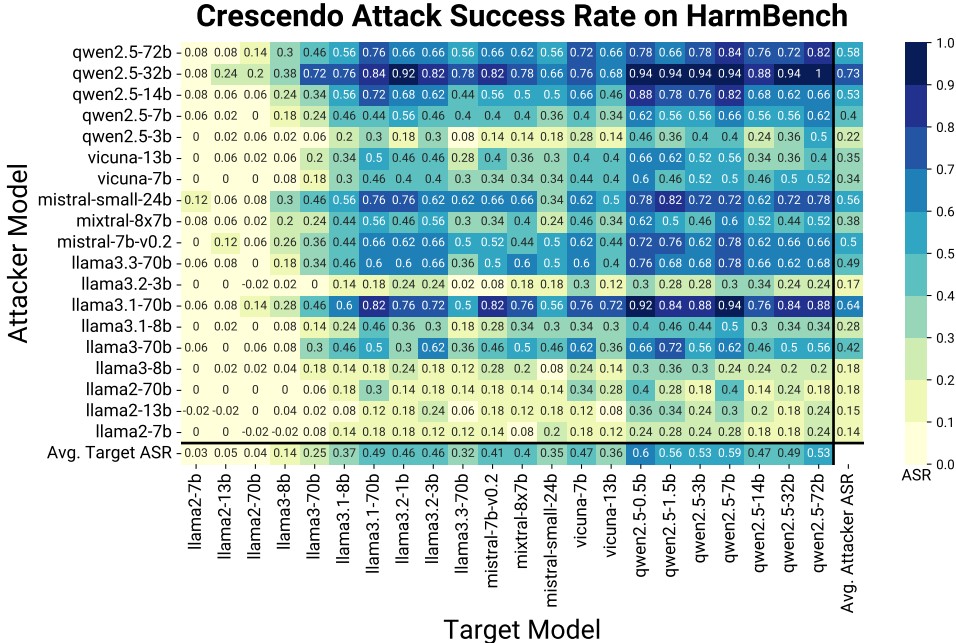

Figure 10: **Attacker-Target Combinations for Crescendo.** Each cell represents the Attack Success Rate (ASR) for a specific attacker-target combination, evaluated on the first 50 queries from HarmBench. All models are sorted by model family, and by generation inside a family. As we discuss in Section 6, Crescendo generally underperforms both TAP and PAIR. Due to computational and monetary constraints, we evaluated Crescendo only on a subset of model combinations, excluding least capable attacker models and closed-source targets.

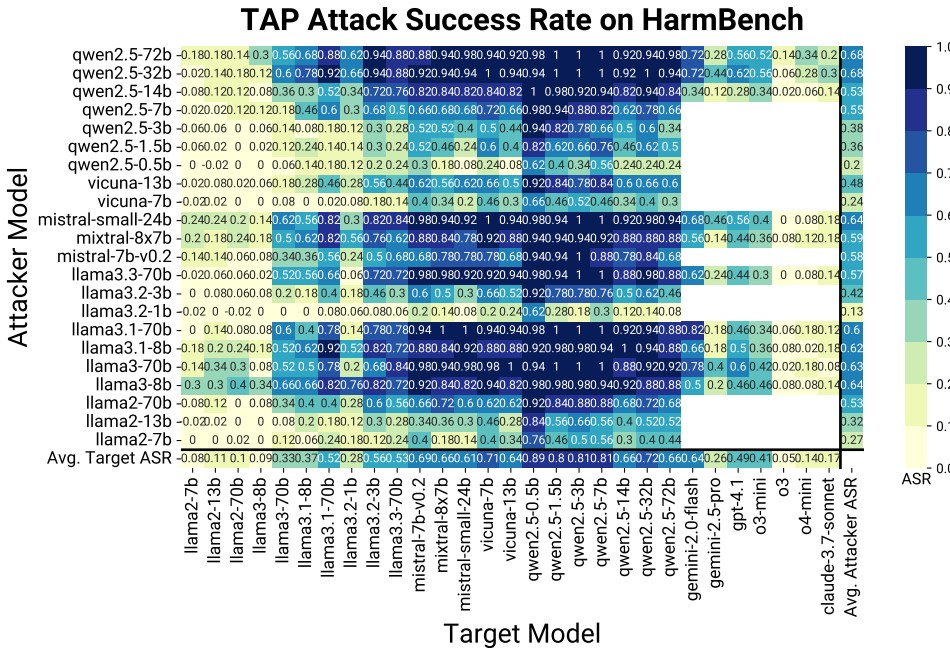

Figure 11: **Attacker-Target Combinations for TAP.** Each cell represents the Attack Success Rate (ASR) for a specific attacker-target combination, evaluated on the first 50 queries from HarmBench. Empty (white) cells represent not evaluated pairs. Due to monetary constraints, against closed-source targets we evaluated only the most capable attacker models.

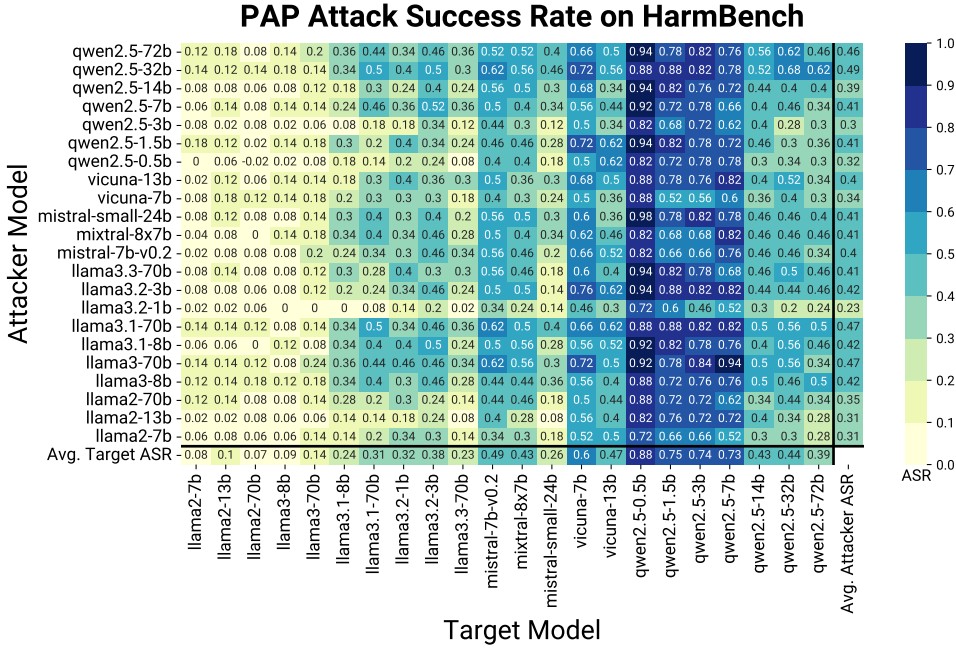

**Figure 12: Attacker-Target Combinations for PAP.** Each cell represents the Attack Success Rate (ASR) for a specific attacker-target combination, evaluated on the first 50 queries from HarmBench. We identify PAP as the least effective attack. Due to computational and monetary constraints, we evaluated PAP only on a subset of model combinations, excluding closed-source targets.

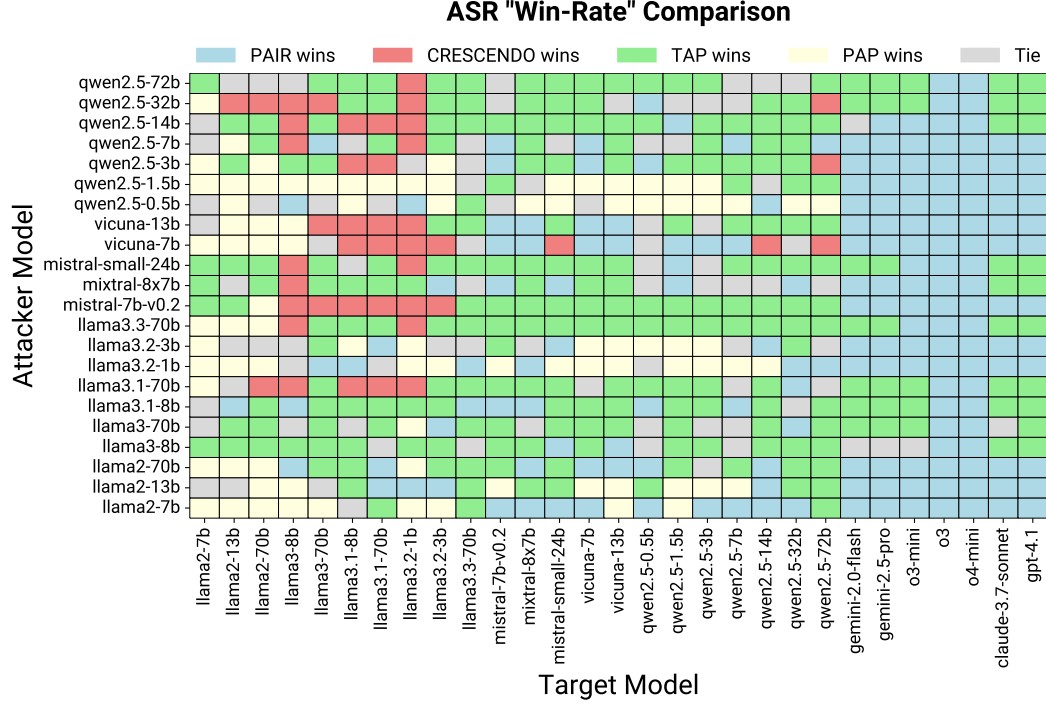

**Figure 13: Attacks "Win-Rate" Comparison for All Attacker-Target Combinations.** Each cell is colored according to the attack method that allowed attacker achieve a higher Attack Success Rate (ASR) against the given target model. A trend emerges, with PAP being more successful against better-safeguarded models. In total, TAP is the best attacking scaffold among considered.

## C    CLEARHARM EVALUATION

In this section, we investigate whether the observed scaling trends generalize to other sources of harmful queries. To this end, we evaluate PAIR on ClearHarm (Hollinsworth et al., 2025; McKenzie et al., 2025), a dataset of unambiguously harmful queries designed to measure ASR on catastrophic risk scenarios. We run PAIR on the first 50 queries and evaluate success using the HarmBench judge.

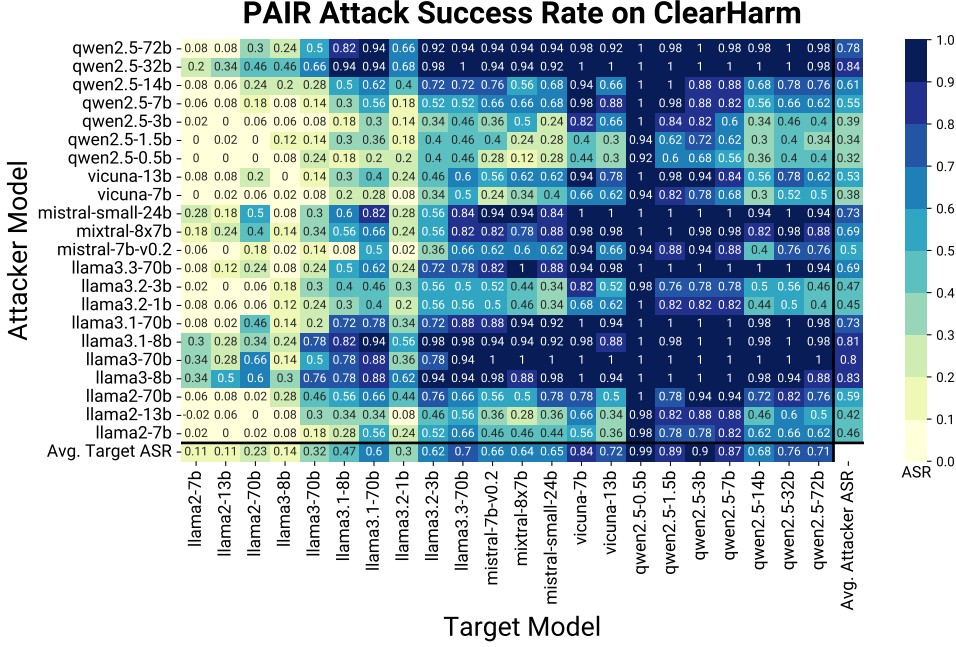

Figure 14: **Attacker-Target Combinations on ClearHarm Dataset.** Each cell represents the Attack Success Rate (ASR) of the PAIR attack for a specific attacker-target combination, evaluated on the first 50 queries from ClearHarm. All attacker models achieve higher ASR compared to the HarmBench dataset (see Figure 9).

Results are shown in Figure 14. All attackers achieve substantially higher ASR on ClearHarm compared to HarmBench, despite ClearHarm queries targeting catastrophic risks. However, the core scaling trends remain robust: ASR scales linearly with attacker capability (Figure 15, $\rho = 0.62$ vs $\rho = 0.67$ on HarmBench), and per-target curves maintain their sigmoid shape (Figure 16). The steeper slopes on ClearHarm mirror the differences between stronger and weaker attacks observed in Figure 8.

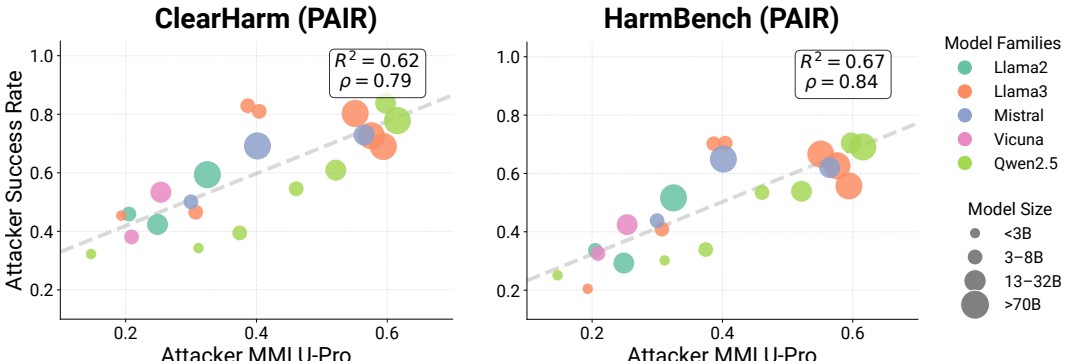

Figure 15: **Comparison of Attacker Average Success Rates between Datasets.** All attacker models achieve higher ASR on the ClearHarm dataset. However, attacker success scales similarly with general attacker capability, following a linear trend across both datasets. Trends plotted on common subset of model pairs of PAIR.

While absolute ASR values are dataset-dependent, the underlying scaling is largely invariant across ClearHarm and HarmBench. Practitioners should evaluate attacks on datasets that reflect their specific threat model or company policy, as dataset choice affects reported success rates but does not alter the fundamental scaling relationships between attacker capability and performance.

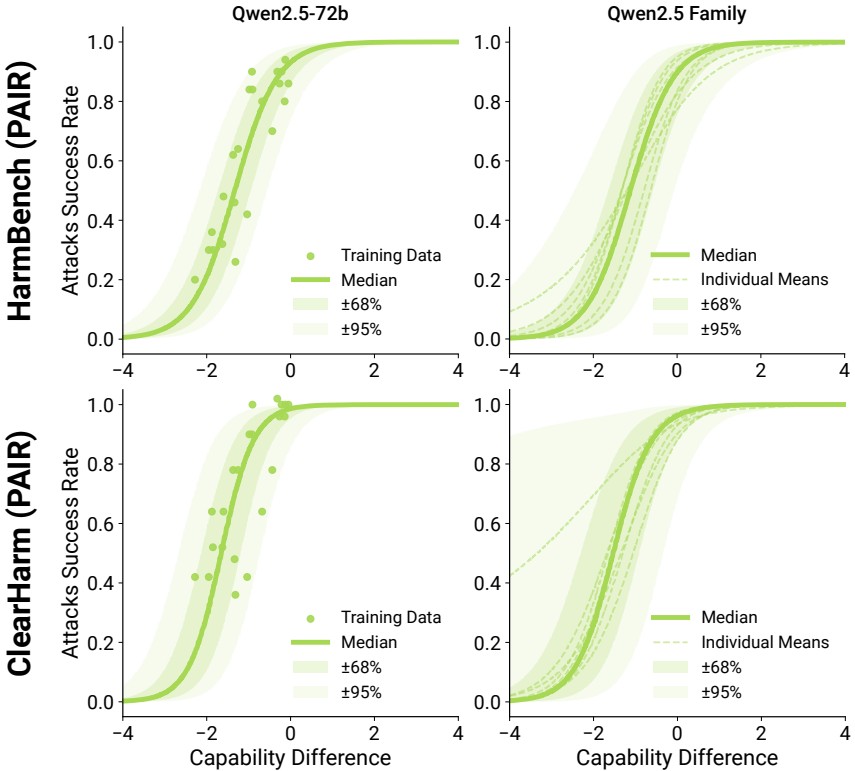

Figure 16: **Comparison of Capability-Based Scaling of PAIR between Datasets.** As ClearHarm emerged as a simpler dataset, the resulting scaling curves are steeper, consistent with the scaling patterns observed between weaker and stronger attacks (e.g., TAP vs. PAIR; see Figure 8).

# D  MODELING DETAILS

In this section, we discuss different modeling approaches and alternative capability gap definitions.

## D.1  PROBLEM SETTING

We aim to model the *attack success rate* (ASR) of jailbreaking attempts as a function of the *capability gap* between the attacker model $\mathcal{A}$ and the target model $\mathcal{T}$. For any given attacker-target pair $a \to t$, our goal is to predict the expected ASR, along with calibrated uncertainty estimates.

To quantify the capability difference , we define the capability gap $\delta_{a \to t}$, as function of MMLU-Pro scores of attacker and target. We compare different capability gap definitions in the following sections. To model worst-case scenario, for the same attacker-target pair we select the highest ASR over considered attacks.

We assume a global non-negative correlation: a weaker attacker (e.g., random token generator) should not outperform a much stronger one (e.g., oracle).

## D.2  PROBLEM FORMALIZATION

Let $\mathcal{D} = \{\mathcal{D}^{(t)}\}_{t=1}^{T}$ denote a collection of $T$ independent datasets. Each dataset $\mathcal{D}^{(t)}$ corresponds to a specific target model $\mathcal{T}^{(t)}$, and contains ASR observations from multiple attacker models. Formally:

$$\mathcal{D}^{(t)} = \{(x_a^{(t)}, y_a^{(t)})\}_{a=1}^{A},$$

where:
- $x_a^{(t)} \in \mathbb{R}$ is the capability gap $\delta$ for the attacker-target pair $a \to t$;

- $y_a^{(t)} = \frac{s_a^{(t)}}{N} \in [0, 1]$ is the observed ASR, where $s_a^{(t)} \in \{0, 1, \dots, N\}$ is the number of successful jailbreaks out of $N$ trials (assumed fixed and known).

Each dataset $\mathcal{D}^{(t)}$ defines a separate regression task. The goal is to infer the predictive distribution for a new input $x_*$:

$$p(y_* \mid x_*, \mathcal{D}^{(t)}),$$

which should capture both the expected ASR and epistemic and aleatoric uncertainties.

For aggregated (per-family) predictive distribution we define a mixture model over targets as follows:

$$p(y_* \mid x_*, \mathcal{D}) = \frac{1}{T} \sum_{t=1}^{T} p(y_* \mid x_*, \mathcal{D}^{(t)}).$$

## D.3  MODELING

We demonstrate in Section 4 that worst-case target vulnerability empirically follows a sigmoid-like curve. For our capability-based predictive model we exploit this observation and apply a logit transformation,

$$\text{logit}(y) = \log\left(\frac{y}{1-y}\right),$$

which maps ASR values $y \in [0, 1]$ to the real line. Consistent with Miller et al. (2021), we then fit a linear model in this new transformed space.

To avoid divergence to $\pm\infty$ when $y = 0$ or $1$, we clip the scores to $\left[\frac{1}{2N}, \frac{2N-1}{2N}\right]$, where $N$ is the number of trials. This clipping is motivated by the fact that the original ASR value is based on $N$ trials, so we use half the resolution of the score to ensure numerical stability while preserving the underlying aleatoric uncertainty.

We specify the linear model as:

$$\text{logit}(y) \sim \mathcal{N}(w \cdot x + b, \sigma),$$

where $x$ denotes the capability gap between attacker and target, and $y$ denotes attack success rate. To accurately infer the predictive distribution we then compare two approaches: Bayesian linear regression and bootstrapped linear regression, as former enables nuanced incorporation of our prior assumptions. We provide our model definitions below.

### D.3.1 BAYESIAN LINEAR REGRESSION

We impose the following priors on the parameters:
- $w \sim \text{HalfNormal}(\sigma_w)$, enforcing a non-negative slope to reflect the assumed non-negative correlation between the capability gap and the ASR;
- $b \sim \mathcal{N}(0, \sigma_b^2)$, allowing symmetric uncertainty around zero for the intercept;
- $\sigma \sim \text{HalfNormal}(\sigma_\sigma)$, ensuring strictly positive observation noise.

The full joint prior is given by:
$$p(w, b, \sigma \mid \sigma_w, \sigma_b, \sigma_\sigma) = \text{HalfNormal}(w \mid \sigma_w) \cdot \mathcal{N}(b \mid 0, \sigma_b^2) \cdot \text{HalfNormal}(\sigma \mid \sigma_\sigma).$$

We define the target-specific hyperparameters as $\Sigma^{(t)} = \left\{ \sigma_w^{(t)}, \sigma_b^{(t)}, \sigma_\sigma^{(t)} \right\}$.

The full posterior distribution, given the data $\mathcal{D}^{(t)}$ and the hyperparameters $\Sigma^{(t)}$, is expressed as:

$$p(w, b, \sigma \mid \mathcal{D}^{(t)}, \Sigma^{(t)}) = \frac{\left[ \prod_{a=1}^{A} \mathcal{N}\left( \text{logit}(y_a^{(t)}) \mid w \cdot x_a^{(t)} + b, \sigma \right) \right] p(w, b, \sigma \mid \Sigma^{(t)})}{\int \left[ \prod_{a=1}^{A} \mathcal{N}\left( \text{logit}(y_a^{(t)}) \mid w \cdot x_a^{(t)} + b, \sigma \right) \right] p(w, b, \sigma \mid \Sigma^{(t)}) \, dw \, db \, d\sigma}.$$

We implement the model using the PyMC python library with the HMC NUTS (No-U-Turn Sampler) algorithm for posterior approximation. Hyperparameters are selected separately for each model via Type II Maximum Likelihood (Empirical Bayes) by maximizing the marginal log likelihood over the range $[0.01, 3.0]$ using Optuna (100 steps). That is, we optimize:

$$\Sigma_*^{(t)} = \arg \max_{\Sigma^{(t)}} \log p(\mathcal{D}^{(t)} \mid \Sigma^{(t)}),$$

where the marginal likelihood is defined as:

$$p(\mathcal{D}^{(t)} \mid \Sigma^{(t)}) = \int \left[ \prod_{a=1}^{A} \mathcal{N}\left( \text{logit}(y_a^{(t)}) \mid w \cdot x_a^{(t)} + b, \sigma \right) \right] p(w, b, \sigma \mid \Sigma^{(t)}) \, dw \, db \, d\sigma.$$

Finally, the predictive distribution for a new observation $y_*$ at a given capability gap $x_*$ is expressed as:

$$p(y_* \mid x_*, \mathcal{D}^{(t)}, \Sigma_*^{(t)}) = \int \text{LogitNormal}(y_* \mid w \cdot x_* + b, \sigma) \, p(w, b, \sigma \mid \mathcal{D}^{(t)}, \Sigma_*^{(t)}) \, dw \, db \, d\sigma.$$

### D.3.2 BOOTSTRAPPED LINEAR REGRESSION

For each target $t$, we generate $N$ bootstrap datasets $\left\{ \mathcal{D}^{(t,n)} \right\}_{n=1}^{N}$ by sampling $A = |\mathcal{D}^{(t)}|$ data points with replacement from the original dataset $\mathcal{D}^{(t)}$.

For each bootstrap dataset $\mathcal{D}^{(t,n)}$ we obtain the maximum likelihood estimates $(w^{(n)}, b^{(n)})$ by solving:

$$(w^{(n)}, b^{(n)}) = \arg \max_{w, b} \prod_{a=1}^{A} \mathcal{N}\left( \text{logit}(y_a^{(t,n)}) \mid w \cdot x_a^{(t,n)} + b, \sigma^2 \right).$$

Then the empirical standard deviation if given by residuals for each bootstrap:

$$\hat{\sigma}^{(n)} = \sqrt{\frac{1}{A} \sum_{a=1}^{A} \left( \text{logit}(y_a^{(t,n)}) - w^{(n)} \cdot x_a^{(t,n)} - b^{(n)} \right)^2}.$$

The final predictive distribution for a new observation $y_*$ at a given capability gap $x_*$ is then approximated as a mixture:

$$p(y_* \mid x_*, \mathcal{D}^{(t)}) = \frac{1}{N} \sum_{n=1}^{N} \mathcal{N}\left( \text{logit}(y_*) \mid w^{(n)} \cdot x_* + b^{(n)}, \hat{\sigma}^{(n)} \right).$$

### D.4 Choosing a Capability-Gap Definition

A capability gap $\delta_{a \to t}$ quantifies how much stronger an attacker $a$ is than a target $t$. Any definition embeds assumptions about how performance differences should scale, especially near the top or bottom of the benchmark range. We evaluate four natural choices, using MMLU-Pro scores as the common capability axis.

**Absolute score gap:** $\quad \delta_{a \to t}^{\text{abs}} = a_{\text{MMLU-Pro}} - t_{\text{MMLU-Pro}}.$

Interpretable, symmetric, and centered at zero. However, it treats the same score difference uniformly across the scale. *Example:* a jump from $0.20 \to 0.30$ (e.g., Vicuna-7b to Mistral-7b) is considered equivalent to a jump from $0.89 \to 0.99$ (e.g., human expert to superhuman model), though the latter may subjectively reflect a more substantial increase in capability.

**Log score ratio:** $\quad \delta_{a \to t}^{\text{log-score}} = \log\big(a_{\text{MMLU-Pro}}/t_{\text{MMLU-Pro}}\big).$

Captures proportional improvements in raw score, but overweights differences at the bottom of the scale. *Example:* the gap between $0.01 \to 0.10$ (incoherent to random guessing) is treated the same as $0.10 \to 1.00$ (random to perfect model), though the latter reflect a far more substantial improvement. Since most current models lie in the lower-mid range, this metric may still perform well empirically.

**Log error ratio:** $\quad \delta_{a \to t}^{\text{log-err}} = \log\big[(1 - t_{\text{MMLU-Pro}})/(1 - a_{\text{MMLU-Pro}})\big].$

Focuses on residual error, which better separates models near the top of the scale. However, like the score ratio, it compresses differences at the lower end. Since most current models lie in the lower-mid range, we expect this metric perform poorly.

**Logit gap:**

$$\delta_{a \to t}^{\text{logit}} = \log\left(\frac{a_{\text{MMLU-Pro}}}{t_{\text{MMLU-Pro}}}\right) + \log\left(\frac{1 - t_{\text{MMLU-Pro}}}{1 - a_{\text{MMLU-Pro}}}\right) = \text{logit}(a_{\text{MMLU-Pro}}) - \text{logit}(t_{\text{MMLU-Pro}}).$$

Combines score- and error-based perspectives into a single, smooth metric. It is symmetric, centred at zero, and better captures variation across the full capability range.

### D.5 Model Comparison

We fit proposed Bootstrapped and Bayesian linear models under each gap definition and assess four criteria: (i) $R^2$ in logit space and (ii) $R^2$ after mapping back to probability, for goodness of fit; (iii) miscoverage at $\alpha = 0.05$, and (iv) Winkler interval score at $\alpha = 0.05$, for predictive uncertainty calibration. We report each metric averaged over all per-target fits (including outliers) in Tab. A.5. For this ablation we used highest achieved ASR over PAIR, Crescendo and Direct Query attacks.

Miscoverage is defined as the proportion of observed ASR values that fall outside the model's predicted 95% confidence interval:

$$\text{Miscoverage} = \frac{1}{n} \sum_{i=1}^{n} \mathbb{I}\left[y_i \notin \widehat{\text{CI}}_{1-\alpha}\right].$$

The Winkler interval score (WIS) (Winkler, 1972) penalizes both miscoverage and overly wide confidence intervals, with the lower score the better.

Among the gap definitions, the absolute and log-error gaps perform noticeably worse across all metrics, for the former suggesting that a linear treatment of capability differences fails to capture the underlying scaling behavior. The log-score and logit gaps perform comparably well, with the log-score showing a marginal advantage. We attribute this to the current lack of models near the upper end of the capability spectrum, which limits the signal that could distinguish logit through residual error scaling. As future models approach this range, we expect the logit-based formulation to better capture improvements near the top of the scale. As both Bayesian and Bootstrapped regressions yield similar scores for predictive uncertainty, we stick to the Bootstrapped version, due to high computational burden of Type-2 MLE. We report per-target fit results in Tab. A.6

**Table A.5: Comparison of Capability Gap Definitions and Regression Methods.** We report average performance across all per-target fits (including outliers), with $\pm$ indicating one standard deviation. Metrics include $R^2$ in logit space (fit quality), $R^2$ after mapping back to probability space, miscoverage and Winkler interval score (both at $\alpha = 0.05$, lower is better). Log-score and logit gaps yield the best fits overall; Bayesian and Bootstrapped regressions yield similar confidence intervals.

| Def. | Reg. | $R^2$ (logit)↑ | $R^2$ (prob)↑ | Avg. Miscoverage↓ | Avg. WIS↓ |
|---|---|---|---|---|---|
| $\delta_{a \to t}^{\text{logit}}$ | Boot. | $0.64 \pm 0.13$ | $0.60 \pm 0.21$ | $0.05 \pm 0.11$ | $0.46 \pm 0.09$ |
| | Bayes | $0.64 \pm 0.12$ | $0.61 \pm 0.21$ | $0.04 \pm 0.11$ | $0.48 \pm 0.10$ |
| $\delta_{a \to t}^{\text{log-score}}$ | Boot. | $0.66 \pm 0.13$ | $0.62 \pm 0.12$ | $0.05 \pm 0.11$ | $0.45 \pm 0.09$ |
| | Bayes | $0.65 \pm 0.14$ | $0.61 \pm 0.20$ | $0.05 \pm 0.11$ | $0.47 \pm 0.09$ |
| $\delta_{a \to t}^{\text{log-err}}$ | Boot. | $0.57 \pm 0.14$ | $0.53 \pm 0.23$ | $0.06 \pm 0.11$ | $0.53 \pm 0.10$ |
| | Bayes | $0.56 \pm 0.14$ | $0.54 \pm 0.21$ | $0.05 \pm 0.11$ | $0.56 \pm 0.12$ |
| $\delta_{a \to t}^{\text{abs}}$ | Boot. | $0.61 \pm 0.14$ | $0.57 \pm 0.22$ | $0.05 \pm 0.10$ | $0.49 \pm 0.09$ |
| | Bayes | $0.59 \pm 0.14$ | $0.58 \pm 0.20$ | $0.04 \pm 0.10$ | $0.53 \pm 0.12$ |

**Table A.6: Per-Target Fits.** Performance of the median bootstrapped regression fit is reported for each target model. For every attacker-target pair we use the maximum ASR achieved across both attacks.

| Target Model Name | $R^2$ (logit) | $R^2$ (prob) | Miscoverage (%) | median $k$ | median $b$ |
|---|---|---|---|---|---|
| Llama-2-7B | 0.50 | 0.16 | 47.8 | 1.18 | -4.15 |
| Llama-2-13B | 0.41 | 0.09 | 17.4 | 1.0 | -3.7 |
| Llama-2-70B | 0.52 | 0.39 | 13.0 | 0.97 | -2.94 |
| Llama-3-8B | 0.63 | 0.56 | 4.3 | 1.23 | -1.78 |
| Llama-3-70B | 0.63 | 0.59 | 4.3 | 1.38 | 0.09 |
| Llama-3.1-8B | 0.77 | 0.75 | 4.3 | 1.45 | -0.43 |
| Llama-3.1-70B | 0.69 | 0.67 | 4.3 | 1.65 | 1.52 |
| Llama-3.2-1B | 0.60 | 0.62 | 4.3 | 1.27 | -1.50 |
| Llama-3.2-3B | 0.71 | 0.71 | 4.3 | 1.40 | -0.16 |
| Llama-3.3-70B | 0.72 | 0.68 | 4.3 | 1.54 | 1.23 |
| Mistral-7B | 0.63 | 0.72 | 4.3 | 1.39 | 0.48 |
| Mixtral-8x7B | 0.72 | 0.75 | 0.0 | 1.80 | 1.33 |
| Mistrall-Small-24B | 0.78 | 0.79 | 4.3 | 1.85 | 1.81 |
| Vicuna-13B | 0.64 | 0.67 | 0.0 | 1.53 | -0.23 |
| Vicuna-7B | 0.81 | 0.80 | 4.3 | 1.42 | 0.16 |
| Qwen-2.5-0.5B | 0.54 | 0.62 | 4.3 | 1.06 | 1.31 |
| Qwen-2.5-1.5B | 0.80 | 0.82 | 13.0 | 2.31 | 1.42 |
| Qwen-2.5-3B | 0.80 | 0.82 | 8.7 | 2.11 | 1.67 |
| Qwen-2.5-7B | 0.78 | 0.80 | 21.7 | 2.15 | 2.80 |
| Qwen-2.5-14B | 0.69 | 0.74 | 0.0 | 1.39 | 1.63 |
| Qwen-2.5-32B | 0.81 | 0.82 | 0.0 | 2.30 | 2.98 |
| Qwen-2.5-72B | 0.73 | 0.79 | 4.3 | 2.05 | 2.83 |
| Gemini-2.0-Flash | 0.76 | 0.68 | 4.3 | 1.93 | 2.23 |
| Gemini-2.5-Pro | 0.47 | 0.31 | 13.0 | 1.18 | 0.30 |
| o3 | 0.47 | 0.29 | 4.3 | 1.01 | 0.73 |
| o3-mini | 0.44 | 0.34 | 0.0 | 0.92 | 0.62 |
| o4-mini | 0.55 | 0.47 | 0.0 | 1.12 | 0.92 |

