# OpenReview forum: "Capability-Based Scaling Trends for LLM-Based Red-Teaming"
_ICLR.cc/2026/Conference — ICLR 2026 Poster_

### Official Review · Reviewer_s6fQ · 2025-10-20

**Soundness:** 2
**Presentation:** 3
**Contribution:** 2
**Rating:** 4
**Confidence:** 3

**Summary:**

This paper investigates how the capability gap between attacker and target language models affects automated jailbreaking success. The central contribution is the discovery of a capability-based scaling law: jailbreaking success (ASR) scales linearly with the capability gap in logit space, with high predictive power across most tested models. The authors demonstrate that this relationship holds across different attack methods (PAIR and Crescendo), though stronger attacks shift the difficulty curve without changing the fundamental slope. The paper also reveals that social science abilities correlate more strongly with jailbreaking effectiveness than STEM capabilities. Additionally, the paper provides methodological insights showing that judge capability affects only prompt selection rather than generation quality, validating cost-saving approaches for red-teaming. Using the discovered scaling law, the authors forecast that human red-teaming will lose effectiveness once models surpass human-level capability.

**Strengths:**

- **Comprehensive empirical evaluation at scale.** The paper tests 22 attacker models against 27 target models across 50 harmful behaviors using two attack methods.

- **Clear quantitative scaling relationship.** The central finding that capability gap predicts attack success with high correlation provides a simple, predictive framework that works well for the majority of tested models.

- **Novel insight about psychological capabilities.** Section 6.1's finding that social science abilities (psychology, philosophy) correlate more strongly with jailbreaking success than STEM capabilities is novel.

- **Transparent treatment of outliers.** Rather than obscuring models that don't fit, the paper explicitly identifies outliers (Llama2 family, Llama3-8B) and acknowledges that heavy safety tuning can decouple defensive capability from general capability.

- **Attack method comparison reveals fundamental invariants.** Section 6.3 shows that different attack methods shift the difficulty curve but preserve the underlying slope, suggesting the scaling relationship reflects fundamental attacker-target dynamics rather than method-specific artifacts.

**Weaknesses:**

The paper acknowledges that 'heavy safety tuning can extend a system's lifespan against stronger attackers', but the framework doesn't incorporate this insight systematically. The scaling law works well for models with no or weak safety alignment, but fails for models with 'exceptional' alignment (Llama2 family, Llama3-8B). This suggests safety alignment strength is an important independent variable that deserves explicit treatment rather than being absorbed into per-family baselines or outlier exclusion. The paper also provides no per-target or per-family scaling curves for frontier models (Claude, GPT, Gemini), which makes it impossible to assess whether heavily-aligned frontier models follow the observed pattern or represent 'exceptional' cases that the framework cannot capture. For forecasting purposes, the key question becomes: will future models follow the observed pattern (Qwen, Mistral) or the 'exceptional' pattern (Llama2)? The paper's forecast implicitly assumes the former but cannot validate this assumption on current frontier models—the very models whose alignment practices are most likely to inform future frontier development.

**Questions:**

Figure 3 (right) appears to show Qwen2.5 models with similar vulnerability across different MMLU-Pro scores, while Figure 4 (bottom left) demonstrates a clear capability-based scaling law for the same family. Can the authors clarify this apparent discrepancy?

---

> ### Author Response · Authors · 2025-11-21
> **Response to Reviewer s6fQ**
>
> Dear Reviewer s6fQ,
>
> We thank you for your thoughtful review, and are pleased that you have found our evaluation comprehensive, and the paper provides novel insights. We are happy to address the concerns you raised below:
>
> > #### Figure 3 (right) appears to show Qwen2.5 models with similar vulnerability across different MMLU-Pro scores, while Figure 4 (bottom left) demonstrates a clear capability-based scaling law for the same family. Can the authors clarify this apparent discrepancy?
>
> Thank you for pointing this out. We are happy to clarify the setup of Figure 3. In Figure 3, we demonstrate the worst-case ASR achieved per target model, so each point is a max over attackers and jailbreak methods given the fixed target model. In our case the strongest model in our suite (Qwen-72b) achieves near perfect ASR >0.92 on all Qwen models including itself. We attribute the small within-family discrepancy (0.08 ASR points) to intrinsic uncertainty.
>
> > #### The paper acknowledges that 'heavy safety tuning can extend a system's lifespan against stronger attackers', but the framework doesn't incorporate this insight systematically. The scaling law works well for models with no or weak safety alignment, but fails for models with 'exceptional' alignment (Llama2 family, Llama3-8B). This suggests safety alignment strength is an important independent variable that deserves explicit treatment rather than being absorbed into per-family baselines or outlier exclusion. The paper also provides no per-target or per-family scaling curves for frontier models (Claude, GPT, Gemini), which makes it impossible to assess whether heavily-aligned frontier models follow the observed pattern or represent 'exceptional' cases that the framework cannot capture.
>
> Let us address the question of how the scaling curves look for the current frontier models. We apologize for any potential confusion, because we do plot per-family (aggregated) scaling for evaluated OpenAI models (gpt-4.1, o3-mini, o4-mini, o3) in grey and Gemini (gemini-2.0-flash, gemini-2.5-pro) in brown in Figure 1 (central part). The trend for Gemini follows closely the Qwen2.5 family, while OpenAI models appear to be more robust. We do find that **all frontier models we evaluate do not follow the ‘very-safe’ scaling curve**, and current alignment efforts can be predicted by the median model scaling laws in our study.
>
>
> > ####  The paper's forecast implicitly assumes the former but cannot validate this assumption on current frontier models—the very models whose alignment practices are most likely to inform future frontier development.
>
> We would like to point out that we do also plot exact ASR values achieved on frontier target models in Figure 4, bottom row (yellow crosses, dubbed as ‘’Test data’’). In every subplot yellow points stay the same and represent how safety of frontier models, that are the most interesting to the community, aligns with open-source families. The curve for the Qwen2.5 family generalizes best for closed-source targets, with "test points" following the same trend as the one aggregated by Qwen2.5 family models, clarifying our point that frontier models fit our trend.
>
>
> > #### For forecasting purposes, the key question becomes: will future models follow the observed pattern (Qwen, Mistral) or the 'exceptional' pattern (Llama2)?
>
>
> That's a great question and very relevant to future development of safe AI. We indeed cannot be sure how ASR would scale for future, unseen, frontier models, as our results indicate that if one were to adopt safety techniques used previously for Llama models, frontier models could be substantially more robust. However, in Figure 4 we see that, in practice, this is not the case and ASR on these models falls predictably as the capability gap increases (in the sense of Figure 3, right). Thus, based on how current frontier models' alignment unfolds, we predict it is following the pattern observed for the Qwen2.5 family.
>
> ---
>
> We thank you for your thorough review, and we hope that we managed to address the concerns you raised. We have updated the manuscript to include evaluation on an additional harmful dataset (ClearHarm, Appendix C) and two additional jailbreaking attacks. We are happy to provide any further clarifications if you have more questions. If you think that your concerns were sufficiently addressed and you think there is merit in our work, we kindly ask you to consider raising your score.

---

> ### Author Response · Authors · 2025-11-27
>
> Dear Reviewer s6fQ,
>
> As the discussion period comes to a close, we would appreciate the opportunity to engage with you regarding our rebuttal.
>
> We wanted to follow up on your comment about frontier models, which are indeed the most important to validate our framework on. As we clarified, the scaling curves for OpenAI and Gemini models are presented in Figure 1 (center), and the achieved ASR on frontier targets appear as yellow "test points" in Figure 4 (bottom row). With Gemini models follow closely follow the Qwen2.5 trend rather than the "very safe" Llama2 trend, which further supports the validity of identified trends for forecasting worst-case safety of frontier LLMs. We have also revised our terminology, using "trends" instead of "laws" throughout the paper, including the title.
>
> We hope you find our response and provided clarifications satisfactory. If so, we would kindly ask you to consider raising your score.
>
> Best regards,
>
> The Authors

---

### Official Review · Reviewer_7ibG · 2025-10-27

**Soundness:** 3
**Presentation:** 4
**Contribution:** 3
**Rating:** 8
**Confidence:** 4

**Summary:**

Summary
The paper tries to create a Scaling law for Jailbreaking of models. It's setup is based off of an harmfully fine-tuned attacker and judge, where the attacker jailbreaks a defender and the judge evaluates success or failure.  They rank the models capabilities based off of their MMLU-Pro scores and then analyze different pairings along with capability vs jailbreak attempts at N attempts and many other combinations.

**Strengths:**

Pros
- Semi-comprehensive testing to try to figure out if there's a scaling law for jailbreaking
- Useful for forecasting

**Weaknesses:**

Cons
- It doesn't seem to highlight limitation that "jailbreaks" might evolve or change forms (imo) namely we might care about different jailbreaks or different sensitive topics in the future.
- I couldn't find if there was a manual QA somewhere. The setup is LLM judges if LLM attacker can jailbreak defender but I'm unsure of false positives or false negatives in this setup.
- The "what makes a good attacker" might be heavily mis-representing the "truth" as models scale they might tend to do better asymetrically on "easier" subjects or questions (like psychology).  This might just be another scaling law plot in disguise. I'd like to see the variance of improved overall performance or the increased compute time variance removed to actually be able to say that humanities capabilities matter more than STEM.

**Questions:**

Notes
- Figure 4
It seems as though the regression was messed up or inaccurately plotted on the llama family data

---

> ### Author Response · Authors · 2025-11-21
> **Response to Reviewer 7ibG (1/2)**
>
> Dear Reviewer 7ibG,
>
> We thank you for your positive review! Let us address the concerns you raised below:
>
> > #### It doesn't seem to highlight limitation that "jailbreaks" might evolve or change forms (imo) namely we might care about different jailbreaks or different sensitive topics in the future.
>
>
> **Based on your suggestion, we now complement our analysis with the ClearHarm dataset [1], which focuses on catastrophic risks. We present these results in Appendix C for a PAIR attack against open-source models.**
>
> Although the ASR achieved on ClearHarm queries is on average higher than on HarmBench,**the identified trends still hold**. In Figure 15 we demonstrate how attackers' average success rate still linearly scales with attackers' general capability (ρ = 0.79). For the per-target fits, the higher baseline ASR on ClearHarm shifts the scaling curve and makes it steeper, mimicking the behaviour for different attacks in Figure 8.
>
> We used ClearHarm to make this new comparison,  given how much focus model providers now put towards CBRN risks, and the uplift malicious actors might get from jailbreaking in "factual domains". We do agree that specific achieved ASR is dependent on the exact evaluated dataset, yet the discovered relative scaling relationship remains the same.
>
>
> > #### I couldn't find if there was a manual QA somewhere. The setup is LLM judges if LLM attacker can jailbreak defender but I'm unsure of false positives or false negatives in this setup.
>
> In this work we use the established HarmBench LLM-judge, first proposed in [2] with demonstrated high human agreement. This judge was thoroughly examined in subsequent literature, e.g. in [3] (Table 4), where the judge has an FPR of 0.04 on HarmBench queries, in [4] (Table 1) FPR of 0.268 on JailbreakBench queries, and high agreement with humans on StrongReject queries in [5] (Table 1).
>
> Since we are using the HarmBench judge with HarmBench queries in pass@25 evaluation, we believe that given the evidence existing in the literature, we are safe to rely on this judge and its discriminative abilities.

---

> > ### Author Response · Authors · 2025-11-21
> > **Response to Reviewer 7ibG (2/2)**
> >
> > > #### The "what makes a good attacker" might be heavily mis-representing the "truth" as models scale they might tend to do better asymmetrically on "easier" subjects or questions (like psychology). This might just be another scaling law plot in disguise. I'd like to see the variance of improved overall performance or the increased compute time variance removed to actually be able to say that humanities capabilities matter more than STEM.
> >
> > To validate this we now expanded our analysis to include Partial Correlations. We calculated the correlation of each benchmark with the Success Rate while controlling for the model's General Capability (proxied by the overall MMLU-Pro score).
> >
> > This isolates the specific contribution of each domain. If success was purely driven by scaling (general capabilities), these partial correlations should effectively vanish. Instead, we observe a distinct divergence where Humanities/Health act as positive drivers.
> >
> > **Table 1.** Pearson and partial correlations between ASR and benchmark.
> >
> > | Benchmark | Category | Raw Correlation (r) | Partial Correlation (r_part) |
> > |-----------|----------|---------------------|------------------------------|
> > | Health (MMLU-Pro) | Health | 0.91 | +0.57 |
> > | Philosophy (MMLU-Pro) | Social Sciences | 0.90 | +0.50 |
> > | Psychology (MMLU-Pro) | Social Sciences | 0.90 | +0.47 |
> > | Other (MMLU-Pro) | Other | 0.90 | +0.45 |
> > | History (MMLU-Pro) | Social Sciences | 0.89 | +0.44 |
> > | Computer Science (MMLU-Pro) | STEM | 0.88 | +0.41 |
> > | Biology (MMLU-Pro) | STEM | 0.89 | +0.35 |
> > | Law (MMLU-Pro) | Social Sciences | 0.87 | +0.20 |
> > | Economics (MMLU-Pro) | Social Sciences | 0.87 | +0.16 |
> > | GSM8K (Strict) | STEM | 0.81 | -0.11 |
> > | IFEval (Inst-Loose) | IFEval | 0.72 | -0.17 |
> > | Engineering (MMLU-Pro) | STEM | 0.82 | -0.18 |
> > | Business (MMLU-Pro) | Social Sciences | 0.85 | -0.35 |
> > | Physics (MMLU-Pro) | STEM | 0.84 | -0.41 |
> > | Chemistry (MMLU-Pro) | STEM | 0.80 | -0.53 |
> > | Math (MMLU-Pro) | STEM | 0.77 | -0.61 |
> >
> > Notably, rigid STEM metrics become *negative* predictors after controlling for general intelligence (r_part from -0.11 to -0.61), while humanities/health remain strong positive predictors (+0.16 to +0.57).
> >
> > While stronger models do have better performance on almost all splits (Table A.2 and A.3), across considered attackers the variance is huge (including on STEM).  We note that the variance in Psychology performance (σ=0.158) is actually lower than in Math (σ=0.170) or GSM8K (σ=0.267), suggesting that Psychology is not systematically 'easier' or scaling faster across our attacker suite. Furthermore, despite Math and Philosophy having similar average scores (0.364 vs 0.387), Philosophy correlates much more strongly with ASR.
> >
> > We emphasize that these correlations do not imply causation (e.g., training on psychology textbooks would create better attackers). Rather, we interpret these results as evidence that models with stronger capabilities in semantic nuance and persuasive communication, which are partly reflected in Social Science knowledge, tend to be more effective at jailbreaking. These capabilities may already be exploited in influence operation attacks, suggesting that the correlation reflects a shared underlying skill set rather than a direct training effect.
> >
> >
> >
> > > #### Figure 4 It seems as though the regression was messed up or inaccurately plotted on the llama family data
> >
> > We believe that you are referring to the Figure 4, bottom, that illustrates aggregated fits within one family. On top of every per-family fit, we plot “test points” which are ASR elicited against closed-source target models. **These points do not correspond to Qwen, Mistral or Llama family**. With these points we wanted to demonstrate how fits for every family generalize to the frontier target, for which we found that the curve for Qwen2.5 generalizes the best. We apologize for the confusion.
> >
> > ---
> >
> > We thank you for your review and positive assessment of our work! We hope that we addressed the questions you raised, and we would be happy to answer any further questions that might arise.
> >
> > ---
> >
> > **References:**
> >
> > [1] https://huggingface.co/datasets/AlignmentResearch/ClearHarm
> >
> > [2] HarmBench: A Standardized Evaluation Framework for Automated Red Teaming and Robust Refusal
> >
> > [3] An Interpretable N-gram Perplexity Threat Model for Large Language Model Jailbreaks
> >
> > [4] JailbreakBench: An Open Robustness Benchmark for Jailbreaking Large Language Models
> >
> > [5] A STRONGREJECT for Empty Jailbreaks

---

> > > ### Comment · Reviewer_7ibG · 2025-11-27
> > >
> > > Thank you for the points. I have read the others reviews and your responses. I would like to maintain my score and encourage the other reviewers to raise their scores in kind.
> > >
> > > I would suggest updating figure 4 for the camera ready (were you to be accepted) to not confuse readers as it possibly takes away from the paper's messaging.

---

> > > > ### Author Response · Authors · 2025-11-27
> > > >
> > > > Dear Reviewer 7ibG,
> > > >
> > > > We sincerely thank you for your time and thoughtful review, and for your support of our work - this means a lot to us!
> > > >
> > > > We agree with your suggestion regarding Figure 4 and will revise it to improve its clarity.
> > > >
> > > > Best regards,
> > > >
> > > > The Authors

---

### Official Review · Reviewer_p1Bo · 2025-10-31

**Soundness:** 3
**Presentation:** 4
**Contribution:** 2
**Rating:** 4
**Confidence:** 4

**Summary:**

This paper investigates automatic red-teaming, in particular what happens if you vary the strength of the attacked and attacker models. The findings show that the strength of the attacker plays a central role in being able to successfully red-team a model. The paper analyzes this from a perspective of MMLU performance of the two models and identifies that the attack success probability follows mildly predictable trends.

**Strengths:**

- I really like the research direction and the work presented in the paper is extensive and comprehensive
- The experiments and approach are innovative with interesting insights which spark many potential ideas for future followups

**Weaknesses:**

- **Scaling Law Misdirection**: My main point of critique is that the paper trying to turn things into a scaling law that really shouldn’t be one. In a way, the notion of there being “scaling laws” is detracting from a lot of the valuable insights this paper is providing. Many curves seem very forced (e.g., in Fig 4) and just drawn through a cloud. I am pretty sure even a linear fit would achieve a similar R^2. There are many parts where correlations are presented as causations, and as discussed in the next point, the experiment rely on a few critical assumptions that may not generalize. As such, I would strongly recommend toning down the "scaling law" claims in this paper.

- **Critical Assumptions**: The paper explores relationships of attacks between one benchmark (MMLU-Pro) and one notion of harmfulness (HarmBench). As such, any claims need to be framed within those terms. By no means are the results generalizable scaling laws. Moreover, since the Llama 2 system prompt is used throughout many of the experiment, it further biases the results since not all models were optimized for this prompt.

- **Explanations are missing**: There are many experiments explaining what is happening, but not many that try to uncover underlying mechanisms at hand. As a result, the results are not immediately actionable other than taking away the advice to "use the best possible model as attacker" which seems rather obvious.

- **Alternative result interpretation**: It could be the case that models with stronger MMLU results also had more safety tuning since they tend to have larger teams behind them. As a result, the key result could just measure the correlation between the post-training effort put into safety and NLU benchmarks.

- **6.1 seems contrived**. While there are correlations between the various MMLU sub-categories, there are a myriad of alternative explanations (e.g., that large models are overfit to STEM categories because of a focus on IMO-style problems). The sections thus also conflates correlation and causation. Moreover, the section has a massive overclaim. Good results in multiple choice tests should absolutely not be categorized as “might rely on psychological insights and persuasiveness” as the authors suggest.

Minor:
- Don’t use white-box vs black-box. These terms have racist connotations.
- The related work section is extremely disappointing. Research existed for longer than 2024.

**Questions:**

n/a

**Details Of Ethics Concerns:**

A version with significant overlap was accepted at a past workshop (https://r2-fm.github.io/). While the workshop claims to not be archival, I can clearly see the paper included in its proceedings: https://openreview.net/forum?id=hQyKnuNg1k — I defer to the ethics review whether this constitutes a violation of policy.

---

> ### Author Response · Authors · 2025-11-21
> **Response to Reviewer p1Bo (1/4)**
>
> Dear Reviewer p1Bo,
>
> We thank you for your review and are glad that you find our work "extensive and comprehensive" and our experiments "innovative" and valuable for the community.
>
> We would like to address the points you raised below:
>
> > #### Scaling laws misdirection. [...] In a way, the notion of there being "scaling laws" is detracting from a lot of the valuable insights this paper is providing. [...] I am pretty sure even a linear fit would achieve a similar R^2.
>
> We appreciate your concern about the "scaling laws" framing and are happy to provide additional clarity on our methodology. In Figure 4 (top row), we present a per-target fit, where we fix one target model and run our suite of attackers against it, using four jailbreaking methods (PAIR, TAP, PAP, Crescendo). We then fit a sigmoid curve from the capability gap between the attacker-target pair to the max ASR achieved in this pair. We chose sigmoid because ASR is a proportion of "solved harmful queries" and is naturally confined between 0 and 1. In short, sigmoid enables meaningful predictions at bigger capability gaps, where linear fit would not.
>
> To better demonstrate this, we performed a comparison of linear fit vs. sigmoid fit and tanh fit in the table below. Note that in Figure 4, for the sake of uncertainty quantification, we bootstrap points and make a prediction with a median fit.
>
> Below we present fits minimizing negative log-likelihood to address your specific concerns.
>
> **Table 1.**  R² for different fits, for the models presented in Figure 4.
> | Target Model | Linear | Sigmoid | Tanh |
> |--------------|--------|---------|------|
> | llama3.2-1b | 0.6109 | 0.6238 | 0.6335 |
> | llama3.2-3b | 0.6992 | 0.7739 | 0.7857 |
> | llama3.1-8b | 0.6856 | 0.7410 | 0.7419 |
> | llama3-8b | 0.4933 | 0.5159 | 0.5290 |
> | llama3-70b | 0.7090 | 0.7323 | 0.7399 |
> | llama3.1-70b | 0.6308 | 0.7181 | 0.7222 |
> | llama3.3-70b | 0.7470 | 0.7723 | 0.7823 |
> | mistral-small-24b | 0.7549 | 0.8074 | 0.8133 |
> | mistral-7b-v0.2 | 0.7405 | 0.8269 | 0.8269 |
> | qwen2.5-0.5b | 0.3987 | 0.7891 | 0.8615 |
> | qwen2.5-1.5b | 0.5225 | 0.7861 | 0.8349 |
> | qwen2.5-3b | 0.5230 | 0.7269 | 0.7505 |
> | qwen2.5-7b | 0.5885 | 0.7254 | 0.7423 |
> | qwen2.5-14b | 0.6682 | 0.7903 | 0.7977 |
> | qwen2.5-32b | 0.7443 | 0.8680 | 0.8867 |
> | qwen2.5-72b | 0.7703 | 0.8497 | 0.8616 |
>
>
> **Table 2.**  Aggregated R² for all per-target fits.
> | Model | Overall R² (pooled over all predictions) | Average R² ± Std (over per-target fits) |
> |-------|-------------------------------------------|----------------------------------------|
> | linear | 0.8694 | 0.6038 ± 0.1582 |
> | sigmoid | 0.9032 | 0.6886 ± 0.1383 |
> | tanh | 0.9058 | 0.7098 ± 0.1350 |
>
> **These results demonstrate that both tanh and sigmoid significantly outperform linear fit.**
>
> > #### Many curves seem very forced (e.g., in Fig 4) and just drawn through a cloud.  [...] As such, I would strongly recommend toning down the "scaling law" claims in this paper.
>
> For Figure 4 bottom, we aggregate per-target fits within one family and plot yellow "test" points on top that correspond to attack runs against closed-source OpenAI and Gemini models. Note that we do not fit curves to these yellow points. It is the same "cloud" applied to every per-family scaling curve, which demonstrates that closed-source models follow the trend identified by the Qwen2.5 family.
>
> We also want to emphasize that the linear trend of average attacker ASR and worst-case target ASR does enable meaningful predictions, given a fixed suite of targets or how current alignment unfolds for a set of attackers. We acknowledge that the term "scaling law" is strongly associated with seminal work such as [1], which demonstrated robust scaling relationships requiring very few inputs to predict expected loss values. While we think that our methodology provides a valuable framework for quantifying deployment risk, **we are happy to find a middle ground and tone down "scaling law" or refrain to a more neutral term such as "scaling trend." Please let us know if you would find this more appropriate.**
>
> > #### The paper explores relationships of attacks between one benchmark (MMLU-Pro)
>
> We additionally measured instruction-following capabilities with IFEval and math capabilities with GSM8k, and found that MMLU-Pro approximates both attacking and defending capabilities much better than these alternatives. However, as we discuss in our limitations section, a model with significantly better attacking skills yet poor MMLU-Pro performance would violate our established scaling trends. Notably, none of the analyzed models exhibited this property.

---

> ### Author Response · Authors · 2025-11-21
> **Response to Reviewer p1Bo (2/4)**
>
> > #### The paper explores [...] one notion of harmfulness (HarmBench). As such, any claims need to be framed within those terms. By no means are the results generalizable scaling laws.
>
> While HarmBench is an established benchmark that spans diverse harm categories, *we now complement our analysis with the ClearHarm dataset [2]*, which focuses on catastrophic risks. We present these results in Appendix C for a PAIR attack against open-source models.
>
> Although the ASR achieved on ClearHarm queries is on average higher than on HarmBench, **the identified trends still hold**. In Figure 15, we demonstrate how attackers' average success rate still linearly scales with attackers' general capability (ρ = 0.79). For the per-target fits, the higher baseline ASR on ClearHarm shifts the scaling curve and makes it steeper, mimicking the behavior observed for different attacks in Figure 8.
>
>
> > #### Moreover, since the Llama 2 system prompt is used throughout many of the experiments, it further biases the results since not all models were optimized for this prompt.
>
> The usage of the same system prompt is precisely what allows us to evaluate models on equal footing, as under the same system prompt or "constitution," they share a common notion of harmfulness. As we discuss in the paper (lines 205-208), it is known that some models are better aligned to region-specific queries (e.g., Qwen often reasons about Chinese law), while others are not, or are not safety-aligned in the first place (e.g., Vicuna). Usage of the same system prompt allows us to account for this. More generally, open-source models are optimized to follow any system prompt, including safety-related prompts. Thus, we believe our approach is fair and leads to a meaningful comparison between models.
>
> Safe system prompts are known to improve model robustness across *different* model families [4], particularly the Llama 2 safe system prompt [3, 6]. For comparison, one can refer to Table 1 of Boreiko et al. [3] and the "standard behaviors" table on page 27 of the HarmBench paper [6]. For example, the GCG attack against Starling-alpha achieves 89% ASR without a safe system prompt but only 61% with it. For Vicuna 13B, GCG achieves 87% ASR without a safe system prompt and 70% with it. Moreover, the authors of [3] explicitly mention that Gemma-7b benefits the most from the safe system prompt, emerging as the safest model with lower elicited ASR than Llama 2 itself.
>
>
> > #### Explanations are missing: There are many experiments explaining what is happening, but not many that try to uncover underlying mechanisms at hand. As a result, the results are not immediately actionable other than taking away the advice to "use the best possible model as attacker" which seems rather obvious.
>
> We provide actionable takeaways at the end of Section 4 (lines 310-315) and at the end of Section 5 (lines 373-377), as well as implications for providers and the jailbreaking community (lines 472-480).
>
> We believe our paper provides valuable insights to both model providers and the jailbreaking community. Most notably, in Section 6.2 (Figure 7), we demonstrate that the Judge model brings no contribution to the attacking capability of the attacker model, and the choice of the judge model plays no role when evaluating ASR@N. This finding addresses high API costs, as red-teamers can abandon expensive closed-source judges without sacrificing attacking pipeline effectiveness for ASR@N.
>
> Please let us know if there are any specific results or aspects you would like us to clarify further.
>
>
> > #### Alternative result interpretation: It could be the case that models with stronger MMLU results also had more safety tuning since they tend to have larger teams behind them. As a result, the key result could just measure the correlation between the post-training effort put into safety and NLU benchmarks.
>
> Let us consider this interpretation. If models with stronger MMLU results consistently had more safety post-training, we would expect to observe linear scaling across all target models with a negative slope in Figure 3 (right). However, we observe a sigmoid relationship (R² = 0.83, ρ = -0.71), where negative linear scaling only emerges beyond a certain capability threshold. This occurs precisely when the target surpasses the attacker, at which point ASR falls rapidly.
>
> Furthermore, when examining frontier model families (OpenAI and Gemini), their scaling coincides with that of Qwen2.5 (Figure 1, middle). This demonstrates that final ASR is not driven exclusively by the amount of alignment or by attacker potency alone, but rather by the interplay between the two, which we capture through our capability gap framework.

---

> ### Author Response · Authors · 2025-11-21
> **Response to Reviewer p1Bo (3/4)**
>
> > #### 6.1 seems contrived. While there are correlations between the various MMLU sub-categories, there are a myriad of alternative explanations (e.g., that large models are overfit to STEM categories because of a focus on IMO-style problems). The section thus also conflates correlation and causation. Moreover, the section has a massive overclaim. Good results in multiple choice tests should absolutely not be categorized as "might rely on psychological insights and persuasiveness" as the authors suggest.
>
> We are happy to complement the analysis of Section 6 with average attacker accuracy on the benchmarks:
>
> **Table 3.** Average model performance on evaluated benchmarks
>
> | Benchmark | Mean ± Std |
> |-----------|------------|
> | GSM8K | 0.600 ± 0.267 |
> | Biology (MMLU-Pro) | 0.600 ± 0.166 |
> | Psychology (MMLU-Pro) | 0.549 ± 0.158 |
> | IFEval (inst-loose) | 0.536 ± 0.145 |
> | Economics (MMLU-Pro) | 0.515 ± 0.164 |
> | Health (MMLU-Pro) | 0.439 ± 0.176 |
> | Other (MMLU-Pro) | 0.430 ± 0.158 |
> | History (MMLU-Pro) | 0.404 ± 0.161 |
> | Computer Science (MMLU-Pro) | 0.401 ± 0.169 |
> | Business (MMLU-Pro) | 0.394 ± 0.171 |
> | Philosophy (MMLU-Pro) | 0.387 ± 0.146 |
> | MMLU-Pro (overall exact-match) | 0.376 ± 0.159 |
> | Math (MMLU-Pro) | 0.364 ± 0.170 |
> | Physics (MMLU-Pro) | 0.335 ± 0.160 |
> | Engineering (MMLU-Pro) | 0.276 ± 0.124 |
> | Chemistry (MMLU-Pro) | 0.273 ± 0.144 |
> | Law (MMLU-Pro) | 0.265 ± 0.106 |
>
> The alternative explanation you provide does not coincide with our empirical observations: Math and Philosophy have similar average scores (0.364 vs 0.387), yet Philosophy correlates much more strongly with ASR. Furthermore, the variance in Psychology performance (σ = 0.158) is actually lower than in Math (σ = 0.170) or GSM8K (σ = 0.267), suggesting that Psychology is not systematically easier or scaling faster across our attacker suite.
>
> We additionally conducted a partial correlation analysis, controlling for general capability of the model as measured by MMLU-Pro. It reveals that Health, Philosophy, and Psychology remain the strongest drivers among all categories.
>
> **Table 4.** Pearson and partial correlations between ASR and benchmark
>
> | Benchmark | Category | Raw Correlation (r) | Partial Correlation (r_part) |
> |-----------|----------|---------------------|------------------------------|
> | Health (MMLU-Pro) | Health | 0.91 | +0.57 |
> | Philosophy (MMLU-Pro) | Social Sciences | 0.90 | +0.50 |
> | Psychology (MMLU-Pro) | Social Sciences | 0.90 | +0.47 |
> | Other (MMLU-Pro) | Other | 0.90 | +0.45 |
> | History (MMLU-Pro) | Social Sciences | 0.89 | +0.44 |
> | Computer Science (MMLU-Pro) | STEM | 0.88 | +0.41 |
> | Biology (MMLU-Pro) | STEM | 0.89 | +0.35 |
> | Law (MMLU-Pro) | Social Sciences | 0.87 | +0.20 |
> | Economics (MMLU-Pro) | Social Sciences | 0.87 | +0.16 |
> | GSM8K (Strict) | STEM | 0.81 | -0.11 |
> | IFEval (Inst-Loose) | IFEval | 0.72 | -0.17 |
> | Engineering (MMLU-Pro) | STEM | 0.82 | -0.18 |
> | Business (MMLU-Pro) | Social Sciences | 0.85 | -0.35 |
> | Physics (MMLU-Pro) | STEM | 0.84 | -0.41 |
> | Chemistry (MMLU-Pro) | STEM | 0.80 | -0.53 |
> | Math (MMLU-Pro) | STEM | 0.77 | -0.61 |
>
> While we agree that MCQ is an insufficient measure of psychological and manipulative abilities, we think that it is obvious that models with strong psychological capabilities can be deployed for influence operations, election manipulation, and social engineering, with recent studies demonstrating concerning real-world applications [7]. We see our work as additional evidence from jailbreaking field, that supports the need for better evaluation and control of such capabilities.
>
> We want to emphasize that while we do not make causal claims beyond reporting the correlations in Section 6.1, **we have now softened the wording to address your concerns.**

---

> > ### Author Response · Authors · 2025-11-21
> > **Response to Reviewer p1Bo (4/4)**
> >
> > > #### Don’t use white-box vs black-box. These terms have racist connotations.
> >
> > These terms are long-established in engineering, security, and ML research, and have no relation to race or ethnicity. In this context, white-box simply denotes that the adversary does have full access to a system's internals, while black-box denotes that it does not.
> >
> > An excerpt from the Inclusive Naming Initiative (https://inclusivenaming.org/word-lists/no-change/blackbox/#:~:text=This%20term%20may%20be%20used,does%20not%20promote%20racial%20bias) supports this usage:
> >
> > "This term may be used without restriction. Blackbox refers to opacity, such as details that aren't visible or are not the focus. This term is not based on a good/bad binary where white is represented as good or black is represented as bad and so does not promote racial bias."
> >
> >
> >
> > > #### The related work section is extremely disappointing. Research existed for longer than 2024.
> >
> > We cite numerous foundational papers on LLMs safety and security from before 2024, including seminal works such as [8, 9, 10, 11]. If there is any particular paper you find relevant that we have missed, we would be happy to add it. Our literature review is in line with well-regarded surveys of the field, for example, such as the ICML 2025 tutorial on "Jailbreaking LLMs and Agentic Systems: Attacks, Defenses, and Evaluations". We encourage you to review slide 68 of the tutorial at https://jailbreak-tutorial.github.io/files/jailbreaking-tutorial.pdf, which illustrates the dynamics of the relatively new field of LLM jailbreaking, with 0 jailbreaking papers existed prior to 2022.
> >
> > ---
> >
> > We thank you for your thoughtful review and are happy to engage further if you have additional questions. We hope that our additional experimental results, which include evaluation on an additional harmful dataset, two additional jailbreaking attacks, and comprehensive correlation analysis, have sufficiently addressed your concerns. If so, we would kindly ask you to consider raising your score.
> >
> > ---
> >
> > **References:**
> >
> > [1] Kaplan et al., Scaling Laws for Neural Language Models. 2020
> >
> > [2] https://huggingface.co/datasets/AlignmentResearch/ClearHarm
> >
> > [3] Boreiko et al. An Interpretable N-gram Perplexity Threat Model for Large Language Model Jailbreaks. 2025.
> >
> > [4] Xu et al. Bag of Tricks: Benchmarking of Jailbreak Attacks on LLMs. 2024.
> >
> > [5] Samvelyan et al. Rainbow Teaming: Open-Ended Generation of Diverse Adversarial Prompts. 2024.
> >
> > [6] Mazeika et al. HarmBench: A Standardized Evaluation Framework for Automated Red Teaming and Robust Refusal. 2024.
> >
> > [7] https://www.science.org/content/article/unethical-ai-research-reddit-under-fire
> >
> > [8] Perez et al. Red teaming language models with language models. 2022.
> >
> > [9] Zou et al. Universal and transferable adversarial attacks on aligned language models. 2023.
> >
> > [10] Yang et al. Shadow alignment: The ease of subverting safely-aligned language models. 2023.
> >
> > [11] Carlini et al. Are aligned neural networks adversarially aligned? 2023.

---

### Official Review · Reviewer_SJAM · 2025-11-02

**Soundness:** 2
**Presentation:** 2
**Contribution:** 2
**Rating:** 4
**Confidence:** 3

**Summary:**

This paper presents a large-scale empirical study on the dynamics of LLM-based red-teaming, reframing the problem from one of absolute model capabilities to one of a relative "capability gap" between the attacker and the target. By evaluating over 600 attacker-target pairs with two distinct LLM-based jailbreak methods (PAIR and Crescendo), the authors establish a "jailbreaking scaling law." This law demonstrates that ASR is not linear but follows a predictable sigmoid function of the capability gap, which they measure using MMLU-Pro benchmark scores. Key findings include that more capable models are both better attackers and more robust targets, and that success in red-teaming correlates more strongly with social science and persuasion-related capabilities than with STEM knowledge. Ultimately, the work forecasts that as AI models become more capable, the effectiveness of any fixed-capability attacker, including human red-teamers, will inevitably and predictably decline.

**Strengths:**

- The analysis is built upon "more than 600 attacker-target combinations", providing a robust empirical basis for its conclusions. The heatmap in Figure 2 visualizes this extensive dataset, lending significant weight to the observed trends and the derived scaling law.

- By fine-tuning models to remove their safety guardrails, they "eliminate the attacker's refusal as a confounding factor". This allows the study to focus on a model's raw ability to craft adversarial prompts, rather than its willingness to engage in the red-teaming task, which is a critical distinction for a clean analysis.

- The "Capability-Based Jailbreaking Scaling Law" (Figure 4) models ASR as a sigmoid function of the capability gap. This provides a concrete, predictive tool for forecasting the future efficacy of red-teaming efforts and reasoning about the security of deployed systems, which is a significant contribution over merely reporting ASR metrics.

**Weaknesses:**

- The paper's central narrative is built on the foundational assumption that general academic benchmark performance (MMLU-Pro) is a valid proxy for the highly specific skills of both offensive jailbreaking and robust defense. This assumption is questionable and introduces a potential circularity. The authors first use MMLU-Pro to define the "capability gap" axis and then, in Section 6.1, show that ASR correlates strongly with the social-science splits of MMLU-Pro (Figure 6). While this correlation is an interesting finding, using MMLU-Pro as the primary independent variable from the outset presumes it is the correct measure. The skills for jailbreaking, such as creative deception, exploiting logical loopholes, and social engineering, are not directly measured by multiple-choice questions. The paper's own data points to this flaw: it dismisses the "early Llama models (Llama2 and Llama3-8b)" as "outliers" (Figure 3) because their robustness is higher than their MMLU-Pro score would predict. These are not mere outliers; they are significant counter-examples suggesting that MMLU-Pro is an incomplete, and at times incorrect, proxy for defensive capability, and that specific safety alignment procedures can break the proposed scaling relationship.

- The paper presents the "unlocking" procedure as a clean removal of safety alignment to reveal a model's intrinsic attacking ability. However, the methodology of fine-tuning on harmful datasets like "BadLlama" and "Shadow Alignment" does not simply revert a model to a neutral base state; it actively trains it to become proficient at generating harmful content in a particular style. This introduces a critical confounder: the experiment may be selecting for models that are better at adapting to this specific fine-tuning task rather than measuring a general, latent "attacking capability." This is evidenced by the authors' own observation of an "unwanted unlocking artifact" where models "overfit to harmful content in the red-teaming prompt". While mitigated with instruction-following data, this core issue remains. The "unlocked models" are fundamentally different from the original ones, meaning the capability scores on the x-axis of Figures 3 and 4 belong to a different model than the one whose attacking performance is being plotted, undermining the directness of the claimed relationship.

**Questions:**

None

---

> ### Author Response · Authors · 2025-11-21
> **Response to Reviewer SJAM (1/2)**
>
> Dear Reviewer SJAM,
>
> We thank you for your thorough and thoughtful review! We are happy to address concerns you pointed out below:
>
> > #### The paper's central narrative is built on the foundational assumption that general academic benchmark performance (MMLU-Pro) is a valid proxy for the highly specific skills of both offensive jailbreaking and robust defense. This assumption is questionable [...] The paper's own data points to this flaw: it dismisses the "early Llama models (Llama2 and Llama3-8b)" as "outliers" (Figure 3) because their robustness is higher than their MMLU-Pro score would predict. These are not mere outliers; they are significant counter-examples suggesting that MMLU-Pro is an incomplete, and at times incorrect, proxy for defensive capability, and that specific safety alignment procedures can break the proposed scaling relationship.
>
> We are in full agreement with the reviewer that MMLU-Pro is a proxy for the underlying attacking/defending capability. We discuss this on lines 319-323, and later on lines 351-353. We do highlight there that the old Llama models follow a different trendline of “safe LLMs”, but this separate trend does not predict the safety of current-generation frontier models, which are predicted by our trends derived from current open-source models, such as Qwen.
>
> However, in our work we demonstrate that MMLU-Pro is a highly predictive proxy that correlates with attack both defense capability that holds surprisingly well over the population of 25 target models we study and characterizes ASR well based on the attacker-target capability gap even for closed-source models (Figure 4, test points).
>
> We believe that the key insight is that a capability-based framework (which can be different fromMMLU-Pro, and better proxy can be slotted in) provides a remarkably simple yet highly predictive framework for predicting jailbreak robustness. Despite all the sophisticated safety techniques, like constitutional AI or deliberative alignment, ASR scales predictably across 25 out of 29 evaluated target models.
>
> > #### This assumption is questionable and introduces a potential circularity. The authors first use MMLU-Pro to define the "capability gap" axis and then, in Section 6.1, show that ASR correlates strongly with the social-science splits of MMLU-Pro (Figure 6). While this correlation is an interesting finding, using MMLU-Pro as the primary independent variable from the outset presumes it is the correct measure. [...]
>
> **There is no circularity between Section 6 and previous sections.** For jailbreaking attacks we run all attacker-target combinations of the models, which allows us to measure average attack success rate per attacker. We then correlate it to benchmark splits, and find that humanities lead to higher correlation numbers. At no point does the choice of MMLU-Pro as the metric for calculating capability gap (or the capability gap itself) influence the analysis in Section 6.
>
> Section 6.1 actually serves as a *validation* of MMLU-Pro's relevance: it shows that the components of MMLU-Pro that correlate most with ASR are precisely those related to persuasion and social sciences, the skills one would intuitively expect to matter for jailbreaking.
>
> > The skills for jailbreaking, such as creative deception, exploiting logical loopholes, and social engineering, are not directly measured by multiple-choice questions.
>
> We agree that MMLU-Pro (or even its social science splits) cannot possibly measure a model's creative deception or social engineering capability directly, and these might be even better ASR predictors. Yet MMLU-Pro is clearly a sufficient proxy for estimating attacker ASR on a fixed suite of targets.
>
> Based on our experiments, we find these a convincing evidence in support of this theory:
>
> - After unlocking models, there are no outliers in Figure 3 (left), with average attacker ASR clearly scaling with attacker capability (R² = 0.76, ρ = 0.86) across all attacks
> - **We additionally evaluated PAIR attack on the ClearHarm (Appendix C) dataset, and linear scaling holds (ρ = 0.79 for PAIR on ClearHarm vs ρ = 0.84 for PAIR on HarmBench)**
>
> If the reviewer would like to suggest a specific benchmark that better captures persuasion, social engineering, or might be interesting otherwise, we would be happy to incorporate it into our analysis.

---

> > ### Author Response · Authors · 2025-11-21
> > **Response to Reviewer SJAM (2/2)**
> >
> > > #### The paper presents the "unlocking" procedure as a clean removal of safety alignment to reveal a model's intrinsic attacking ability. However, the methodology of fine-tuning on harmful datasets like "BadLlama" and "Shadow Alignment" does not simply revert a model to a neutral base state; it actively trains it to become proficient at generating harmful content in a particular style. This introduces a critical confounder: the experiment may be selecting for models that are better at adapting to this specific fine-tuning task rather than measuring a general, latent "attacking capability." This is evidenced by the authors' own observation of an "unwanted unlocking artifact" where models "overfit to harmful content in the red-teaming prompt". While mitigated with instruction-following data, this core issue remains. The "unlocked models" are fundamentally different from the original ones, meaning the capability scores on the x-axis of Figures 3 and 4 belong to a different model than the one whose attacking performance is being plotted, undermining the directness of the claimed relationship.
> >
> > Overall, we agree that this is a fair concern. However, we do not think that it undermines the main findings of the paper and the discovered scaling relationships for two reasons:
> > 1) Every attacker undergoes nearly identical unlocking, for which we verify that: (1) models do not refuse harmful queries post-unlocking, and (2) general capabilities are preserved (Table A.2 shows no apparent degradation in benchmark scores post-unlocking).
> >
> >     **We always use the post-unlocking benchmark scores when plotting capability.** The capability scores belong to the exact same model checkpoint whose attacking performance we measure. Thus, for Figure 3 (left), MMLU-Pro scores of unlocked models are used. For Figure 4, when calculating the capability gap for Qwen2.5-72B attacking Qwen2.5-72B, the gap is never exactly 0, as we always use the attacker's unlocked vs original target’s score for the gap calculation.
> >
> >     While it's theoretically possible that our unlocking procedure favors certain model architectures (or sizes) over others, we see no clear empirical evidence for this, except for the three smallest models in our analysis. For the rest, the procedure works consistently across diverse model families and architectures.
> >
> > 2) The unlocking procedure does not train models to be better at adversarial attack generation specifically. The harmful datasets are essentially instruction-response pairs that teach models to comply with harmful requests (and a request to jailbreak a target model is indeed harmful). The attacking capability we measure comes from the model's underlying abilities, not from specialized training on attack generation.
> >
> >     If one considers our unlocked models as "attackers conditioned on this specific unlocking procedure" rather than revealing pure intrinsic capability, our findings remain valid: they characterize how attackers (created via a standard, reproducible procedure) scale with capability, and how capability gaps predict success regardless of the specific elicitation method used.
> >
> > ---
> > We believe our framework provides a valuable first-order model of scaling dynamics between attacker and target LLMs. The consistency of our results across hundreds of model pairs, multiple jailbreak methods, diverse model families, and two distinct harmful datasets (HarmBench and ClearHarm, which we added in the revision) suggests we are capturing real and important phenomena.  **Based on your feedback, we have added additional discussion of MMLU-Pro limitations in Section 6.**
> >
> > We hope that the provided clarifications and additional experiments have addressed your concerns, and we kindly ask you to consider raising your score.

---

> > ### Comment · Reviewer_SJAM · 2025-11-22
> >
> > I thank the authors for their detailed rebuttal.
> >
> > Regarding the point on circularity: I accept your clarification that the measurement of ASR is procedurally independent of the MMLU-Pro scoring. "Circularity" may have been the wrong terminology to describe the issue.
> >
> > However, the substance of my critique remains: **Construct Validity**.
> >
> > The paper posits a "Scaling Law" based on the premise that general capability (MMLU-Pro) is a valid proxy for both attacking competence and defensive robustness. While Section 6.1 validates MMLU-Pro as a proxy for *attacking* capability, the exclusion of the Llama-2 and Llama-3 families as "outliers" demonstrates that MMLU-Pro is a fundamentally flawed proxy for *defensive* capability.
> >
> > A "law" that fails to predict the robustness of the most prominent open-weight safety-tuned models suggests that defensive robustness is not primarily a function of general capability, but of specific safety alignment techniques. By excluding the models that disprove the trend, the paper derives a relationship that holds only for models where safety training is either absent or ineffective.
> >
> > Consequently, while the measurements are not circular, the conclusion is brittle. The strong correlation relies on filtering out the data points (heavy safety tuning) that matter most for real-world red-teaming, undermining the claim of a generalizable "jailbreaking scaling law."

---

> > > ### Author Response · Authors · 2025-11-22
> > > **Authors Reply to Reviewer SJAM**
> > >
> > > We appreciate the reviewer's continued engagement with our work. Let us clarify concerns raised in the last comment.
> > >
> > > > the exclusion of the Llama-2 and Llama-3 families
> > >
> > > This appears to be a misunderstanding. **We do not exclude the Llama-3 family.** Of the 29 target models we evaluate:
> > > - Only 4 models deviate from the trend in Figure 3 (right) and follow a "safer" trend: Llama-2-7B, Llama-2-13B, Llama-2-70B, and Llama-3-8B
> > > - The remaining **25 models, including Llama-3-70B, Llama-3.1-8B, Llama-3.1-70B, and other open-source and closed-source models, follow the same scaling in the sense of Figure 3 (right)**
> > >
> > > > MMLU-Pro is a fundamentally flawed proxy for defensive capability
> > >
> > > We explicitly acknowledge that it is only a proxy on lines 322-323. Then, based on your comment, in the Discussion section (lines 466-471), we acknowledge that there should be better proxies for defending capabilities, like FLOPs spent on alignment, yet we do not have access to them. In Section 3.1, we explicitly introduce that due to incomparability of different alignment approaches, our framework accounts for this by focusing on per-target scaling laws rather than claiming a one universal relationship for any target models. **We are happy to further revisit our phrasing and tone down "scaling law" component if the reviewer would find it more appropriate.**
> > >
> > > > the paper derives a relationship that holds only for models where safety training is either absent or ineffective. [...] The strong correlation relies on filtering out the data points (heavy safety tuning) that matter most for real-world red-teaming, undermining the claim of a generalizable "jailbreaking scaling law"
> > >
> > > **Our discovered scaling trend predicts worst-case ASR on current frontier models: Claude-3.7-Sonnet, Gemini-2.0-Flash, Gemini-2.5-Pro, o3, o3-mini, and o4-mini.**
> > >
> > > These are the one of the most capable and heavily safety-tuned models that exhist, with large dedicated safety teams. The fact that our capability-based framework predicts their robustness demonstrates its relevance for precisely the real-world scenarios that matter most.
> > >
> > > **The 4 outlier models (Llama-2 family and Llama-3-8B) represent older, less relevant safety approaches.** Meanwhile, all current-generation open-source models (Qwen2.5, **all later Llamas**, Mistral, Vicuna) and frontier closed-source models follow our predicted trends. If anything, this suggests that modern safety training does not fundamentally break capability-based scaling.
> > >
> > > Our contribution is demonstrating that despite new sophisticated safety techniques (constitutional AI, RLHF, deliberative alignment), capability gaps remain highly predictive of jailbreaking success across the models that define the current threat landscape. This is precisely what makes our findings actionable for real-world red-teaming and AI safety.
> > >
> > > ---
> > >
> > > We hope this clarification resolves the misunderstanding, and we kindly ask the reviewer to reconsider their assessment in light of what the paper claims and demonstrates.

---

> > > > ### Comment · Reviewer_SJAM · 2025-11-22
> > > >
> > > > I strongly encourage you to adopt your proposed change to **'tone down the scaling law component.'**
> > > >
> > > > While you argue that the Llama-2 family and Llama-3-8B are 'older' or 'less relevant,' Llama-3-8B is a recent, widely deployed model (released 2024). The fact that such a prominent model must be treated as an outlier suggests that specific safety fine-tuning methods can indeed break the capability-based scaling relationship.
> > > >
> > > > A 'Law' implies universality. The existence of significant counter-examples, even if labeled 'outliers', demonstrates that this is an **empirical trend** contingent on current safety paradigms, not an immutable law.
> > > >
> > > > If the paper is revised to frame this as a 'Capability-Based Scaling Trend' and explicitly discusses *why* certain safety interventions (like those in Llama-3-8B) can defy this trend, I would be willing to raise my score.

---

> > > > > ### Author Response · Authors · 2025-11-23
> > > > > **Authors Reply to Reviewer SJAM**
> > > > >
> > > > > Dear Reviewer SJAM,
> > > > >
> > > > > We agree with your characterization that "Law" rather implies universality and that the presence of significant counter-examples demonstrates this is an empirical trend rather than an immutable law.
> > > > >
> > > > > **We have now revisited our paper and abstained from using the term "Scaling Law."** We now refer to the identified relationships as "trends" and "curves" when describing the functions fitted to the capability gap. We have updated the figures and removed mentions of "scaling law" throughout the paper.
> > > > >
> > > > > We have also updated the paper title to **"Capability-Based Scaling Trends for LLM-Based Red-Teaming"**; while we cannot update the title on OpenReview at this stage of the review process, we will do so in the camera-ready version if the paper is accepted. We explicitly acknowledge that the earliest Llama models constitute outliers and limitations of MMLU-Pro throughout Sections 4 and 5 (lines 229-231, lines 340-348, lines 361-364).
> > > > >
> > > > > In the revised version, we have additionally **significantly expanded the limitations discussion in Section 7** (lines 473-485), where we now offer explanations for why Llama2 and Llama3-8b violate the trend and discuss what better proxies might look like.
> > > > >
> > > > > - - -
> > > > >
> > > > > We once again thank you for your openness to discussion, engagement with our paper, and dedication to helping us improve our work. We hope that the current revision addresses your concerns regarding framing and welcome any further suggestions. We would greatly appreciate if you would consider raising your score.

---

> > > > > > ### Comment · Reviewer_SJAM · 2025-11-24
> > > > > >
> > > > > > Thank you. I have raised the score.

---

> ### Author Response · Authors · 2025-11-27
>
> Dear Reviewer SJAM,
>
> We are grateful for your engagement during the rebuttal period, your support of our work, and **raising the score from 4 to 6.**
>
> Thank you for helping us improve it!
>
> Best regards,
>
> The Authors

---

### Author Response · Authors · 2025-11-30
**Final Remarks**

We sincerely thank the reviewers for their time, detailed feedback, and constructive engagement throughout the review process.

We appreciate the recognition of our work’s extensive scope and insights for the jailbreaking community. **Reviewer SJAM** noted that the work provides a "robust empirical basis" and that the capability-based formulation offers a "concrete, predictive tool... which is a significant contribution." **Reviewer p1Bo** highlighted that the experiments are "innovative with interesting insights," while **Reviewer s6fQ** liked the "comprehensive empirical evaluation at scale" and the "novel insight" regarding psychological capabilities. Moreover, during the limited discussion period, **Reviewer SJAM** increased their score from 4 to 6, while **Reviewer 7ibG** maintained their score of 8 and encouraged the other reviewers to raise their scores.

Below, we summarize the key changes and improvements made to the manuscript during the rebuttal phase:

1. **Refined Framing: "Scaling Trends"**

    Based on the feedback from Reviewer SJAM and Reviewer p1Bo, we revised the paper's framing and our terminology. **We have updated the title to "Capability-Based Scaling Trends for LLM-Based Red-Teaming".** We significantly expanded the discussion regarding outliers, updated the figures, and made our claims more precise by now classifying them as 'trends' and ‘curves’ rather than 'laws.' This re-frames our findings as a robust empirical model for risk assessment rather than a universal "law". **Reviewer SJAM agreed to increase the score from 4 to 6 after these changes.**

2. **Expanded Dataset Coverage: ClearHarm Evaluation**

    To address concerns regarding the generalizability of discovered trends across different harm types, we complemented our evaluation with the **ClearHarm** dataset, which focuses on potentially catastrophic risks, such as CBRN. As detailed in Appendix C, while absolute attack success rates shift, the scaling trends remain consistent (ρ=0.79), confirming that our findings hold across different definitions of harm.

3. **Expanded Attack Suite**

    To improve the generalizability of our claims, we expanded our evaluation to include two additional LLM-based jailbreak attacks, TAP and PAP. These experiments confirm that stronger attack methods shift the scaling curve but do not alter the discovered capability-based scaling relationship.

4. **Correlation Analysis**

    To address concerns regarding Section 6.1, we provided a partial correlation analysis. By controlling for general capability (MMLU-Pro), we isolated the specific contribution of different domains. The results show that STEM metrics become negative predictors, while Social Sciences remain positive predictors. We have made our writing of the implications of these findings more precise: We find that there is a correlation, even after controlling for capability, but we now highlight that performance on MMLU-Pro/Psychology should not be understood as equivalent to generic persuasiveness, which is harder to evaluate with current benchmarks.

---

We believe our capability-based framework offers providers and red-teamers an actionable tool for risk assessment. Further, our work provides valuable insights for the research community, such as demonstrating that expensive judge models are unnecessary for effective ASR@N evaluation, enabling significantly cheaper pipelines, and highlighting the need for persuasiveness evaluations beyond standard STEM benchmarks.

We have now addressed all major concerns raised by the reviewers. We thank the reviewers for their guidance in refining this work, though we regret that the premature end of the discussion period prevented some reviewers from responding.

---

### Meta-Review · Area_Chair_ixXQ · 2026-01-05

**Summary:**

This paper studies LLM-based red-teaming through the lens of the capability gap between an attacker and a target model, and hypothesizes that attack success rate (ASR) can be predicted from the models’ MMLU-Pro scores. Through extensive experiments over many attacker–target pairs, the authors report that ASR exhibits a consistent empirical relationship with this MMLU-Pro–based capability gap, and fit a “jailbreaking scaling curve” intended to predict attack success from the gap. The paper further reports that ASR correlates particularly strongly with performance on the social-science splits of MMLU-Pro.

**Reviewer Concerns:**

Reviewer SJAM
- Validity of MMLU-Pro as a capability prox): Partially resolved at the framing level, but not substantively resolved. The reviewer questions whether MMLU-Pro is an appropriate proxy for the skills underlying both attacking and defensive robustness. They point to notable counterexamples—most prominently the Llama-2 family and Llama-3-8B—whose robustness does not align with what their MMLU-Pro scores would predict. The authors acknowledge this limitation and revised the paper to avoid “scaling law” language, reframing the claims as empirical “scaling trends/curves,” but they do not provide a stronger proxy or a mechanism that explains (or models) these outliers.
- Potential circularity in using MMLU-Pro score: Resolved. There was a misunderstanding in the evaluation procedure and the authors clarified it.
- Unlocking may act as a confounder: Not fully resolved: the authors acknowledge the concern and argue that the main empirical relationships remain informative under a standardized unlocking procedure, but the potential confounding effect is not definitively ruled out.

Reviewer p1Bo
- Scaling law misdirection: Partially addressed but not fully resolved. “Even a linear model would be a good fit” claim has been resolved by the additional results, but the fundamental concern on the “scaling law” remains.
- Scope limitations (MMLU-Pro + HarmBench; generalizability): The reviewer notes the study is anchored to a particular capability proxy (MMLU-Pro) and a particular harmfulness definition/dataset (HarmBench), and therefore should not be presented as a broadly generalizable law. Partially addressed: the authors added experiments on the ClearHarm dataset (catastrophic-risk focused) and argued that the qualitative trends persist, but the overall framework remains tied to specific benchmarks, prompts, and evaluation choices.
- Explanations are missing: Partially addressed. The reviewer argues the paper offers limited mechanistic insight and the main actionable takeaway risks collapsing to the obvious “use the strongest attacker.”  The authors highlight practical implications (e.g., judge model choice/cost tradeoffs for ASR@N pipelines) and expand discussion of implications, but deeper causal/mechanistic explanations remain limited.
- The results may be measuring the correlation between post-training effort for safety and NLU benchmarks: Partially resolved. The authors argue that the observed shape (e.g., non-linear drop beyond a threshold) and cross-family comparisons are more consistent with an interaction captured by capability gap than with a single monotone “more capability → more safety” explanation, but the paper does not directly measure or control for alignment/post-training effort as an independent variable.

Reviewer 7ibG
- Jailbreak might evolve or change its forms in the future: Partially resolved. The authors supplemented results on ClearHarm dataset including catastrophic risks.
- Reliability of LLM judge / lack of manual QA: Partially resolved. The reviewer asks for evidence about false positives/negatives (ideally via manual auditing). The authors do not provide new human QA, but respond by citing prior validation results for the HarmBench judge (human agreement / reported error rates) and argue this is sufficient for their evaluation setting.
- “Social science → better attacker” may be a misguided interpretation due to asymmetric scaling: Resolved. The authors add a partial-correlation analysis controlling for overall capability (MMLU-Pro) and claim humanities/health remain positive predictors while several STEM splits turn negative.

Reviewer s6fQ
- The safety alignment strength itself is an important independent variable: Not resolved.
- No per-target or per-family scaling curves for frontier models: Partially resolved. The authors clarify that they do include aggregated (per-family) curves for some frontier families (e.g., OpenAI and Gemini). However, this is still more limited than the open-weight setting (i.e., it does not fully replicate the same per-target/per-attacker sweep for frontier models).

**Reviewer Scores:**

Except for the most positive reviewer (7biG), the remaining reviews stayed in the borderline range, and none appeared willing to actively champion the paper. The primary reason for rejection is that—by the authors’ own acknowledgement—the proposed “scaling law” is not a universal law but an empirical trend contingent on the specific proxy (MMLU-Pro) and the evaluated setup.

The presence of notable counterexamples demonstrates that defensive robustness is substantially influenced by the intensity of safety alignment. As the authors admit, MMLU-Pro is an "insufficient measure" for defensive capability in these cases. Consequently, I am not convinced the proposed scaling trend is reliable enough to serve as a forecasting tool for attack success, as it appears to break down precisely when strong safety measures are applied.

Moreover, even granting the reported empirical trend, the paper currently offers limited actionable implications. The primary takeaway risks being interpreted as a relatively expected conclusion—namely, that stronger models tend to yield stronger attackers—without yet providing a deeper explanation of the underlying mechanism or clear guidance on how such correlations might inform the design of more robust defenses. I view this primarily as an opportunity for future work rather than a fundamental flaw.

That said, I agree with the reviewers that this submission makes a valuable empirical contribution. In particular, the unusually large-scale and systematic experimental study is carefully executed and provides a useful empirical reference point that will likely be of interest to the red-teaming and AI safety communities. While the technical novelty is modest and the mechanistic understanding remains limited, the breadth of the empirical findings help establish a solid baseline and highlight important empirical regularities that merit further investigation.

---

### Decision · Program_Chairs · 2026-01-26

Accept (Poster)